# The thioesterase APT1 is a bidirectional-adjustment redox sensor

Tuo Ji[1], Lihua Zheng[1], Jiale Wu[1], Mei Duan [1], Qianwen Liu[1], Peng Liu[1], Chen Shen [1], Jinling Liu[1], Qinyi Ye[1], Jiangqi Wen[2,3], Jiangli Dong [1,4] ✉ & Tao Wang [1,4] ✉

The adjustment of cellular redox homeostasis is essential in when responding to environmental perturbations, and the mechanism by which cells distinguish between normal and oxidized states through sensors is also important. In this study, we found that *acyl-protein thioesterase 1* (*APT1*) is a redox sensor. Under normal physiological conditions, APT1 exists as a monomer through *S*-glutathionylation at C20, C22 and C37, which inhibits its enzymatic activity. Under oxidative conditions, APT1 senses the oxidative signal and is tetramerized, which makes it functional. Tetrameric APT1 depalmitoylates S-acetylated NAC (NACsa), and NACsa relocates to the nucleus, increases the cellular glutathione/oxidized glutathione (GSH/GSSG) ratio through the upregulation of *glyoxalase I* expression, and resists oxidative stress. When oxidative stress is alleviated, APT1 is found in monomeric form. Here, we describe a mechanism through which APT1 mediates a fine-tuned and balanced intracellular redox system in plant defence responses to biotic and abiotic stresses and provide insights into the design of stress-resistant crops.

Living organisms constantly confront various stresses and must respond to external threats in a timely manner by adjusting their endogenous survival strategies. Plants cannot move to evade harsh circumstances and have thus evolved an enhanced ability to perceive and resist various biotic and abiotic stresses[1,2]. External stresses trigger oxidative stress under various environmental conditions. Although excessive oxidative stress causes damage to cells and tissues, maintenance of a physiological level of oxidative challenge is essential for governing life processes through redox signalling[3–5], and redox sensors have always been a topic of acute interest in animal and plant stress research[6–12]. Redox sensors recognize potential stress- and injury-related events before large-scale tissue damage occurs and balance the low threshold sensitivity with high fidelity[13]. Studies conducted in recent years have identified a few redox-related sensors in plants, including HPCA1[7] and QSOX1[14], which perceive extracellular $H_2O_2$ and transmit and regulate downstream redox responses through different signalling pathways. However, plant cells must distinguish

normal and oxidized states and subsequently repair cellular damage and regulate redox homeostasis[3,4,15,16]. The maintenance of the redox balance in cells depends on many endogenous substances[17–21]; among these substances, GSH participates in detoxification and ROS removal[4,22–24], and the glutathione/oxidized glutathione (GSH/GSSG) ratio is a typical indicator of the redox balance in plants[17,25] and mammals[26,27]. Nonetheless, the redox sensors in previous studies only showed their ability to sense oxidative signals, and the mechanism cells use to perceive changes in redox molecules under normal and oxidized conditions is not well understood. The sulfhydryl group is one of the key biochemical factors involved in signal transduction and organism adaptation to oxidative stress[28]. Although some studies have indicated that the change in sulfur valence in biochemical processes is correlated with the normal and oxidized states of cells[17,29,30], few results have revealed the detailed molecular mechanisms of sulfhydryl-containing proteins functioning as redox sensors. Different oxidative posttranslational modifications (Ox-PTMs) occurring at the sulfurous

[1]State Key Laboratory of Agrobiotechnology, College of Biological Sciences, China Agricultural University, Beijing 100193, China. [2]Institute for Agricultural Biosciences, Oklahoma State University, Ardmore, OK 73401, USA. [3]Department of Plant and Soil Sciences, Oklahoma State University, Stillwater, OK 74078, USA. [4]These authors contributed equally: Jiangli Dong, Tao Wang. ✉e-mail: dongjl@cau.edu.cn; wangt@cau.edu.cn

group might determine the function of the modified proteins under oxidative stress[24].

Thioesterases are usually functional proteins involved in fatty acid synthesis and depalmitoylation[31,32]. As shown in our previous study, the thioesterase *acyl-protein thioesterase 1* (APT1) could depalmitoylate the transcription factor NACsa (NAC with *S*-acetylation) under drought stress[33]. Under normal physiological conditions, NACsa proteins are anchored on the cytoplasmic membrane of the cell through the palmitoylation of C26. After drought stress-induced depalmitoylation by APT1, NACsa relocates to the nucleus and upregulates the transcription of the downstream target gene *glyoxalase I* (*GLYI*). Upregulated *GLYI* expression leads to increases in the reduced glutathione/oxidized glutathione (GSH/GSSG) ratio[33]. However, the specific signals perceived by APT1 under drought stress are unknown.

Here, we discovered that APT1 is a bidirectionally adjusted redox sensor. Under normal cellular physiological conditions, *S*-glutathionylated APT1 is a monomer. Monomeric APT1 senses the ROS signal and forms a tetramer. Tetrameric APT1 converts ROS signals to transcriptional activation signals through NACsa relocation. NACsa that has relocated to the nucleus upregulates *GLYI* expression, leading to an increase in the GSH/GSSG ratio and a decrease in the intracellular $H_2O_2$ content, resulting in an increased antioxidant capacity of the plant. Upon removal of oxidative stress, APT1 is found in a monomeric form. This type of sensor-regulator redox balance system is conserved in *Glycine max*, *Arabidopsis thaliana* and *Solanum habrochaites*.

## Results

### APT1 accumulates on the cytoplasmic membrane and depalmitoylates NACsa in response to oxidative stress

In a previous study, we found that APT1 regulates the nuclear relocation of NACsa through depalmitoylation and modulates the GSH/GSSG ratio in cells under drought stress[33]. The increase in the $H_2O_2$ content in cells under drought stress implies that APT1 might be directly involved in redox homeostasis. Therefore, we chose 4 mM $H_2O_2$ as the main oxidative stress treatment in this study (the extracellular $H_2O_2$ concentration of *Medicago truncatula* roots under dehydration stress for 2 h was ~4 mM; see Supplementary Fig. 1a and the Methods for details). We expressed *pNACsa:NACsa-GFP* in the presence/absence of *APT1* in *M. truncatula*. After treatment with 4 mM $H_2O_2$ for 30 min, NACsa-GFP relocated to the nucleus in wild-type (WT) ecotype R108 cells. Conversely, NACsa-GFP remained on the cytoplasmic membrane in *apt1* mutant cells after oxidative stress treatment (Fig. 1a). Based on this finding, APT1 is the only regulator of NACsa depalmitoylation in *M. truncatula*, and NACsa relocation might indicate whether APT1 possesses enzymatic activity.

Interestingly, unlike many thioesterases with depalmitoylation activity that are localized in the membrane[32], APT1 is localized in both the cytoplasmic membrane and the cytoplasm in *M. truncatula* protoplasts, regardless of whether APT1 was promoted by CaMV *p35S* or the *MtAPT1* promoter (2.0 kb) (Fig. 1b). We transiently overexpressed *APT1-GFP* in *M. truncatula* hairy roots. The results showed that APT1 was localized in both the cytoplasmic membrane and the cytoplasm under normal conditions and after 4 mM $H_2O_2$ treatment for 30 min or 6 h (Supplementary Fig. 1b, d). We performed a component separation experiment in which we separated the cytoplasm and the cytoplasmic membrane components using *pAPT1:APT1-GFP* transient transgenic *M. truncatula* hairy roots. The immunoblot results indicated that the cytoplasmic membrane fraction obtained after 30 min of treatment with 4 mM $H_2O_2$ contained more APT1-GFP and that the cytoplasm fraction contained less APT1-GFP than the controls (Fig. 1c). Therefore, in the nonstressed state, APT1 is mainly located in the cytoplasm, while oxidative stress promotes APT1 accumulation on the cell membrane, which may improve the opportunity for APT1 to interact with NACsa and thus the release of membrane-anchored NACsa to the nucleus[33].

We generated *pAPT1:GUS* transgenic lines to study the tissue-specific expression of *APT1*. APT1 was expressed at higher levels in the roots (Fig. 1e and i) than in the hypocotyl leaf (Fig. 1e ii), stem (Fig. 1e iii) and leaf (Fig. 1e iv), and similar findings were obtained by immunoblotting of the GUS protein (Supplementary Fig. 1c) and RT–qPCR analyses (Supplementary Fig. 1d) of different tissues. These results indicate that *APT1* is mainly expressed in the root.

### APT1 is a monomer through *S*-glutathionylation of C20, C22 and C37

We generated an APT1 protein fused with a 6 × His tag that was purified in *E. coli* to study the mechanism of APT1 function under oxidative stress and examine the features. Nonreducing SDS–PAGE results showed that APT1 mainly appeared as dimers and tetramers in vitro (Fig. 2a lane 1). APT1 also formed tetramers under drought stress in vivo in *M. truncatula* (Supplementary Fig. 2a). Interestingly, after the application of the reducing agents dithiothreitol (DTT) (1 mM), glutathione (GSH) (10 mM), or cysteine (10 mM) or the application of different concentrations of 2-hydroxy-1-ethanethiol (2-ME) during purification, APT1 appeared as monomers (Fig. 2a, Supplementary Fig. 2c). However, APT1 could not be demultimerized by GSSG, NADPH or $H_2O_2$ (Supplementary Fig. 2b), but $H_2O_2$ could remultimerize the APT1 protein that had been treated with 10 mM GSH (Fig. 2b, Supplementary Fig. 2d). Thus, reducing agents with sulfhydryl groups promote APT1 demultimerization, and $H_2O_2$ reverses this process in vitro. We also examined the effect of different oxidizing/reducing agents on the enzymatic activity of APT1 using 4-nitrophenyl octanoate (4-NPC) as a substrate. The enzymatic activity of APT1 was significantly inhibited by the application of 10 mM GSH, but $H_2O_2$ exerted no significant effect (Supplementary Fig. 2e).

Microscale thermophoresis (MST) and immunoblot assays were conducted to study the mechanism by which GSH inhibits APT1 multimerization and enzymatic activity. GSH but not free cysteines (Cys) in the control group interacted with APT1 (Fig. 2c). The demultimerization of APT1 was positively correlated with the GSH concentration, particularly if the GSH concentration was similar to the normal physiological concentration in cells (Fig. 2d), which was equal to ~1–10 mM in previous studies[34,35]. At the same GSH concentration, decreases in the APT1 protein concentration were associated with a decrease in multimerization and a gradual increase in the monomer level (Fig. 2e). Based on these results, high relative concentrations of GSH lead to the formation of APT1 monomers.

GSH usually interacts with cysteine residues in proteins, and three cysteine residues are present in the APT1 protein: C20, C22, and C37 (Supplementary Fig. 2f). By performing an LC–MS/MS analysis, we showed that C20, C22, and C37 of APT1 were *S*-glutathionylated (Supplementary Fig. 3). Molecular docking was used to illustrate the binding mode of GSH with C20, C22, and C37 of APT1 (Fig. 2g–i, Supplementary Fig. 4), and GSH formed suitable steric complementarity with the binding sites of C20, C22, and C37 in APT1 and formed stable hydrogen bonds with nearby residues. Furthermore, we purified APT1 proteins with different site-directed mutations, including APT1$^{C20S}$, APT1$^{C22S}$, APT1$^{C37S}$, APT1$^{C20SC22S}$, APT1$^{C20SC37S}$, APT1$^{C22SC37S}$ and APT1$^{C20SC22SC37S}$ (APT$^{SSS}$), and then detected the proteins using nonreducing SDS–PAGE. The immunoblot results showed that APT1, APT1$^{C20S}$ and APT1$^{C22S}$ were multimerized; APT1$^{C37S}$, APT1$^{C20SC22S}$, APT1$^{C20SC37S}$, and APT1$^{C22SC37S}$ showed less multimerization; and APT1 with all three cysteines mutated (APT$^{SSS}$) was not multimerized (Fig. 2f). Taken together, the key for APT1 monomerization is the *S*-glutathionylation of the three cysteine residues.

### APT1 transforms from a monomer to a tetramer after sensing ROS

The dynamic change in the APT1 state under oxidative stress was studied in vivo. *p35S:APT1-GFP* was expressed in *apt1* mutant plants,

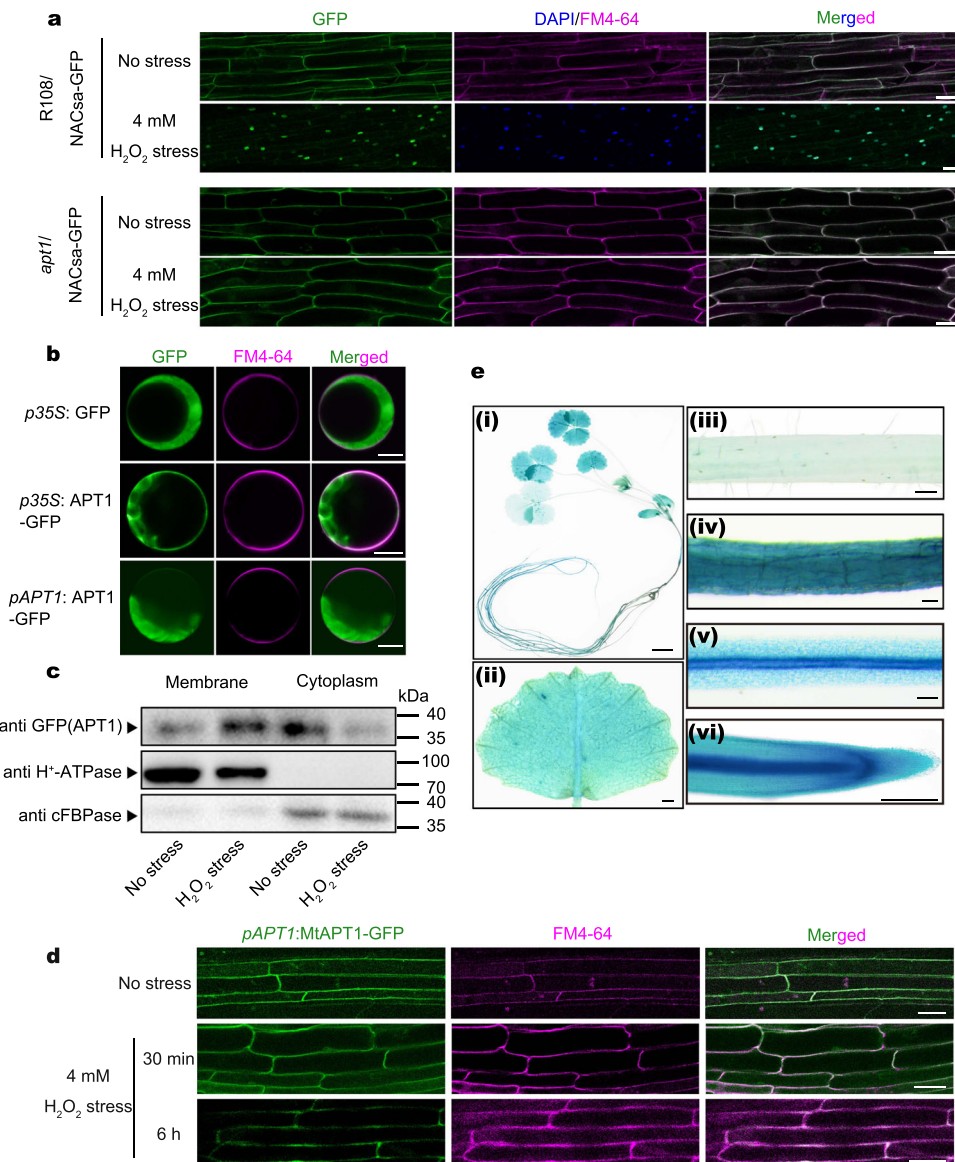

**Fig. 1 | MtAPT1 accumulates on the membrane, regulates NAC deacetylation under oxidative stress and is mainly expressed in roots. a** Confocal images of transgenic *M. truncatula* roots expressing NACsa-GFP fusion proteins driven by the constitutive NACsa promoter in R108 and the *apt1* mutant exposed to normal or 4 mM $H_2O_2$ stress conditions for 30 min. Bars = 20 μm. **b** Confocal images of R108 *M. truncatula* protoplasts expressing APT1-GFP fusion proteins driven by the constitutive CaMV 35 S promoter and APT1 promoter. Bars = 10 μm. **c** Component separation of transient transgenic *M. truncatula* hairy roots expressing APT1-GFP fusion proteins driven by the constitutive APT1 promoter in the *apt1* mutant after exposure to normal or 4 mM $H_2O_2$ stress conditions for 30 min. **d** Confocal images of transient transgenic *M. truncatula* hairy roots expressing APT1-GFP fusion proteins driven by the constitutive APT1 promoter in the *apt1* mutant after exposure to normal or 4 mM $H_2O_2$ stress conditions for 30 min and 6 h. Bars = 20 μm. **e** Detection of tissue-specific expression using X-Gluc staining of various tissues of transgenic *M. truncatula* expressing GUS proteins driven by the constitutive APT1 promoter (2.0 kb), namely, whole plant (i), leaf (ii), stem (iii), hypocotyl (iv), root (v), and root tip (vi). Bars = 1 cm (i), 1 mm (ii), or 0.2 mm (iii, iv, v, and vi). DAPI was used to label the nucleus and excited at a wavelength of 385 nm, and FM4-64 was used to label the cytoplasmic membrane and excited at a wavelength of 546 nm. The source data are provided as a Source Data file.

and nonreducing simple Jess capillary-based electrophoresis and nonreducing SDS–PAGE immunoblot results showed that APT1 transformed from a monomeric state in the absence of stress to a multimeric state after exposure to 4 mM $H_2O_2$ oxidative stress treatment for 15 min to 6 h (Fig. 3a i and Supplementary Fig. 5d i). The APT1 protein level decreased at 6 h and after 6 h (Fig. 3a ii and Supplementary Fig. 5d ii), suggesting that APT1 may be degraded. In a recovery experiment, seedlings were exposed to oxidative stress and then transferred to normal conditions to continue growing; after recovery for 96 h, the tetrameric form of APT1 disappeared, and the protein was present as a monomeric form (Fig. 3b and Supplementary Fig. 5e). Moreover, we utilized a GSH biotinylated analogue, biotinylated ethyl glutathione (BioGEE), which is cell permeable, to detect the *S*-glutathionylation

level of APT1. In seedlings that were treated with 0.1 mM BioGEE for 1 h, subjected to oxidative stress and allowed to recover, the *S*-glutathionylation level of APT1 decreased during exposure to oxidative stress and increased after recovery (Supplementary Fig. 5f). These results indicate that APT1 can sense ROS signals to form tetramers under oxidative conditions.

## C20/C22 and C37 are essential for the thioesterase activity of APT1

We generated different cysteine residue mutants of APT1 fused with red fluorescent protein (RFP) and complemented them in *pNACsa:-NACsa-GFP/apt1* transgenic plants to test whether the cysteines of APT1 were related to its thioesterase activity. NACsa was retained in the

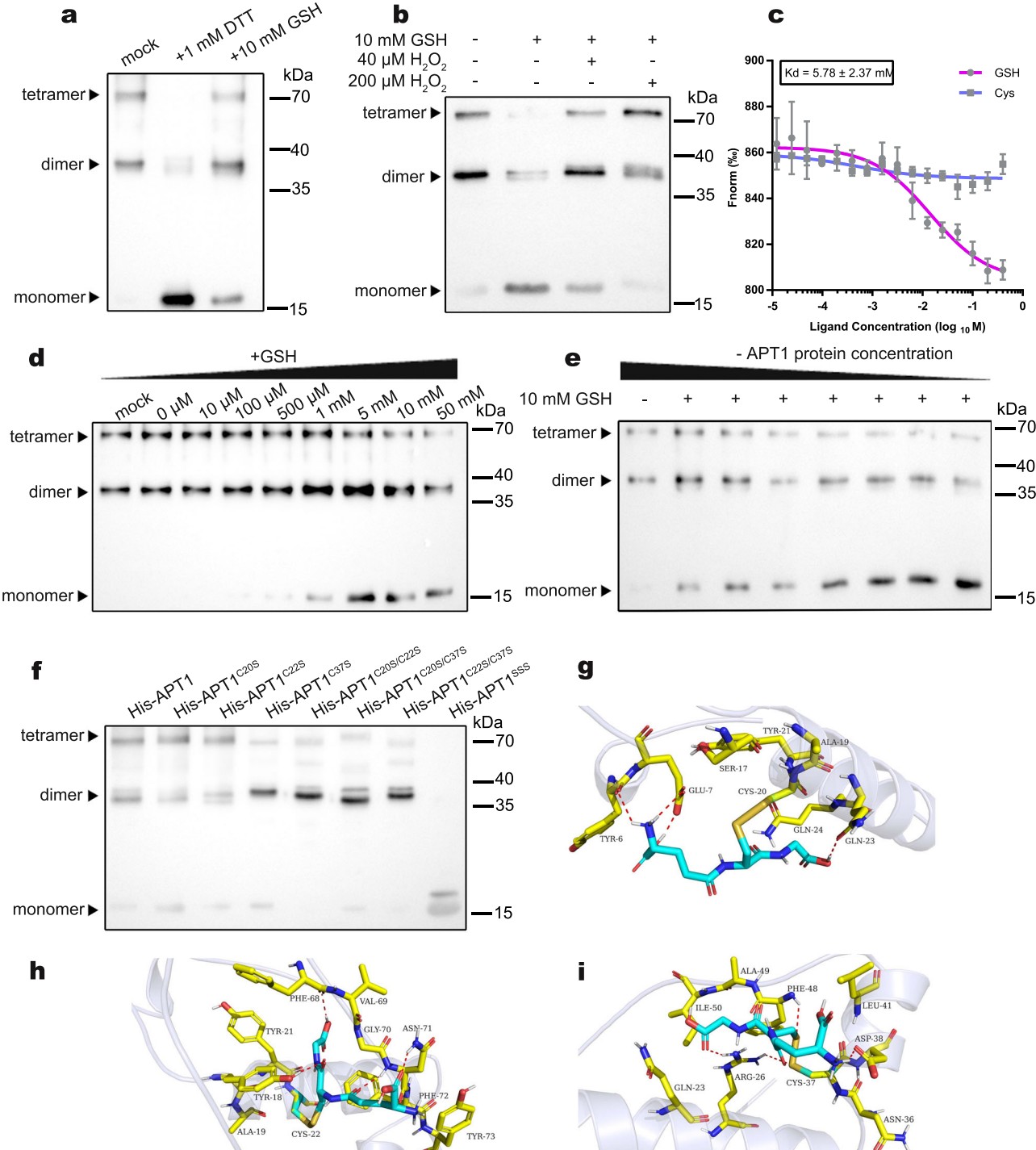

**Fig. 2 | APT1 remains in a monomeric state by interacting with GSH in vitro.**
**a** Western blot of 6 × His-APT1 fusion proteins separated on nonreducing SDS–PAGE gels after purification (left, lane 1). APT1 appeared as monomers after treatment with 1 mM DTT (middle, lane 2) or 10 mM GSH (right, lane 3). **b** Western blot of proteins separated on nonreducing SDS–PAGE gels showed that APT1 (same amount of protein) appeared as monomers after treatment with 10 mM GSH, but further treatment with $H_2O_2$ (40 μM and 200 μM) promoted APT1 remultimerization. **c** The APT1 protein interacts with GSH, as detected using MST. The data are presented as the means and SEs of three independent experiments. **d** Western blot of proteins separated on nonreducing SDS–PAGE gels showed that the same

amount and same concentration of the APT1 protein interacted with the same amount but different concentrations of GSH. **e** Western blot of proteins separated on nonreducing SDS–PAGE gels showed that the same amount (1 ng) of APT1 protein interacted with different volumes but the same concentration (10 mM) of GSH. All different concentrations of APT1 protein mixtures with GSH were loaded into the gel, and the immunoblotted amount of the protein was consistent. **f** Western blot of 6 × His-APT1 fusion protein with different cysteine site mutations separated on nonreducing SDS–PAGE gels after purification. **g–i** Binding mode of GSH with C20 (**g**), C22 (**h**), and C37 (**i**) in APT1, as determined using molecular docking. The source data are provided as a Source Data file.

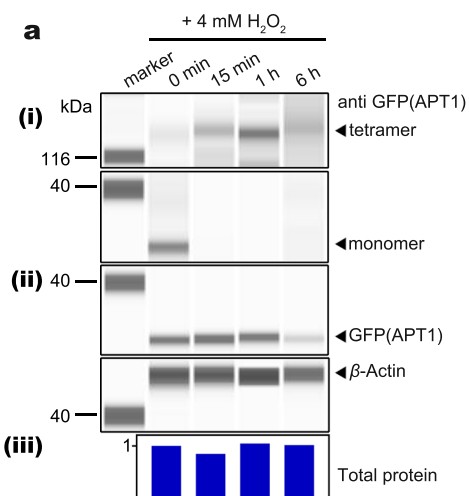

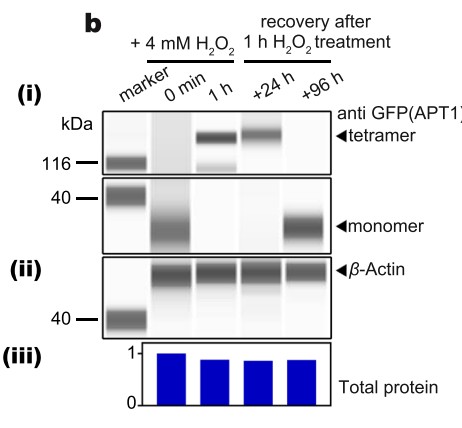

**Fig. 3 | Oxidative stress promotes APT1 tetramer formation in vivo. a** Simple Jess assays of transgenic *M. truncatula apt1* roots showing the multimerization of the overexpressed APT1-GFP fusion protein after treatment with 4 mM H₂O₂; detection was performed with anti-GFP (APT1), nonreducing (i) and reducing (ii) conditions, and with anti-β-actin (control protein) (ii). Total protein was analysed after protein normalization for the quantification of total protein (iii). a i to a iii show results from the same sample with the same loading volume (3 μl). **b** Simple Jess assays of transgenic *M. truncatula apt1* roots showing the multimerization of the over-expressed APT1-GFP fusion protein after treatment with 4 mM H₂O₂ for 1 h and after recovery after 4 mM H₂O₂ treatment; detection was performed with anti-GFP (APT1) (nonreducing) (i) and anti-β-actin (control protein) (ii). Total protein (iii) was analysed after protein normalization for the quantification of total protein. b i to a iii show results from the same sample with the same loading volume (3 μl). The source data are provided as a Source Data file.

cytoplasmic membrane in the nonstressed state, and APT1^C20S and APT1^C22S promoted NACsa-GFP translocation to the nucleus after 4 mM H₂O₂ treatment in vivo and maintained high levels of thioesterase activity, similar to APT1 (Fig. 4a i-iii and 4b i-iii). However, other mutants did not promote NACsa relocation under stress (Fig. 4a iv-viii and 4b iv-viii). This finding indicates the functional redundancy of C20/C22 of APT1 and suggests that the function of APT1 depends on both C20/C22 and C37. We transiently expressed APT1^SSS (C20SC22SC37S) in *apt1* mutant cells, and APT1^SSS was retained in the cytoplasm (Supplementary Fig. 5g) and was unable to form a multimer under nonstress or oxidative stress conditions (Supplementary Fig. 5i). We analysed the conservation of the APT1 sequence in different species, including *Glycine max*, *Arabidopsis thaliana*, *Solanum habrochaites* and *E. coli*, and found that C22 and C37 but not C20 are conserved in these species (Supplementary Fig. 2f). This finding implies that C20/C22 and C37 are needed for the function of thioesterase.

### Multimerized APT1 reduces ROS accumulation by upregulating *GLYI* expression and increasing the GSH/GSSG ratio

We screened a series of homozygous *Tnt1*-insertion mutants in *M. truncatula* with insertions located in different introns and exons to study the function of APT1 and the APT1-NACsa signalling pathway under oxidative stress in vivo (Supplementary Fig. 6a–h): NF15130 (designated *apt1-1*), NF5250 (designated *nacsa-1*), NF9803 (designated *nacsa-2*), NF10049 (designated *glyI-1*) and NF20885 (designated *glyI−2*). We generated the CRISPR/Cas9 *apt1* mutant lines shown in Supplementary Fig. 7a and designated them *apt1-2* and *apt1-3*. We also generated the complementary stable transgenic lines *pAPT1:APT1-3×flag/apt1* and *pAPT1:APT1^SSS−3×flag/apt1* (Supplementary Fig. 7b). Using these transgenic lines, we detected the differences between the *apt1* mutant and the complementary lines under oxidative stress. The fluorescent probes H₂DCFDA and ThiolTracker Violet were used to visualize ROS and GSH in vivo, respectively. After treatment with 4 mM H₂O₂, WT plants and the complementary line *pAPT1:APT1-3×flag/apt1* exhibited significantly lower ROS levels and higher GSH levels in root tips than the *apt1*, *nacsa*, *glyI* and *pAPT1:APT1^SSS−3×flag/apt1* lines. The GSH levels in root tips after recovery from the treatment were also higher in the WT and complementary lines (Fig. 5a, b). A statistical analysis of the mean fluorescence intensity under the same detection conditions also revealed similar results (Fig. 5c–e, h and k). The *glyI* mutant showed higher ROS levels and lower GSH levels in the absence of stress, potentially due to the accumulation of methylglyoxal. After oxidative stress, the WT and complementary lines exhibited a significant upregulation of *GLYI* expression, and this finding was observed both after short-term (15 min) and after long-term (6 h) stress exposure; however, the expression of *GLYI* in *apt1* and *pAPT1:APT1^SSS−3×flag/apt1* was not significantly upregulated (Fig. 5f). The WT and complementary lines maintained a higher GSH/GSSG ratio than the *apt1* and *pAPT1:APT1^SSS−3×flag/apt1* lines (Fig. 5i) after stress, which is beneficial for plants because it reduces the damage caused by ROS bursts under abiotic stress. The results of DAB and NBT staining also supported the conclusions described above: the *apt1-1* and *pAPT1:APT1^SSS−3×flag/apt1* lines exhibited higher ROS accumulation (including H₂O₂ and O₂·) than the WT and complementary lines after treatment with 4 mM H₂O₂ for 15 min or 6 h (Supplementary Fig. 8a). According to these results, APT1 is involved in the scavenging of excessive ROS, increases the GSH/GSSG ratio in cells under oxidative stress and enhances the resistance of plants to oxidative stress.

The use of *apt1*, *nacsa* and *glyI* mutants for the detection of redox changes under stress conditions showed that the WT seedlings exhibited greater upregulation of *GLYI* expression and a higher GSH/GSSG ratio under stress than the mutant strain. DAB and NBT staining also revealed lower accumulation of ROS in the WT seedlings, consistent with the results of the fluorescent probe assay (Fig. 5a, b, g and j, Supplementary Fig. 8b). These results indicated that APT1-NACsa-GLYI functioned as a complete signalling pathway. The nucleophilic sulfhydryl group allows GSH to form GSH conjugates with substances containing electrophilic groups[36], which are often used to relieve the cytotoxicity of electrophilic oxides, such as methylglyoxal[37] and others[38]. Therefore, the increases in *GLYI* expression and GSH levels promoted by tetrameric APT1 reduce cytotoxicity and intracellular ROS accumulation.

### *apt1* mutant seedlings show enhanced resistance to pathogens caused by an increase in intracellular ROS accumulation

Biotic stress is also related to the pressure of an ROS burst, but its effect often hinders the infection efficiency of invaders. Therefore, ROS may be beneficial to seedlings exposed to biotic stress. Three

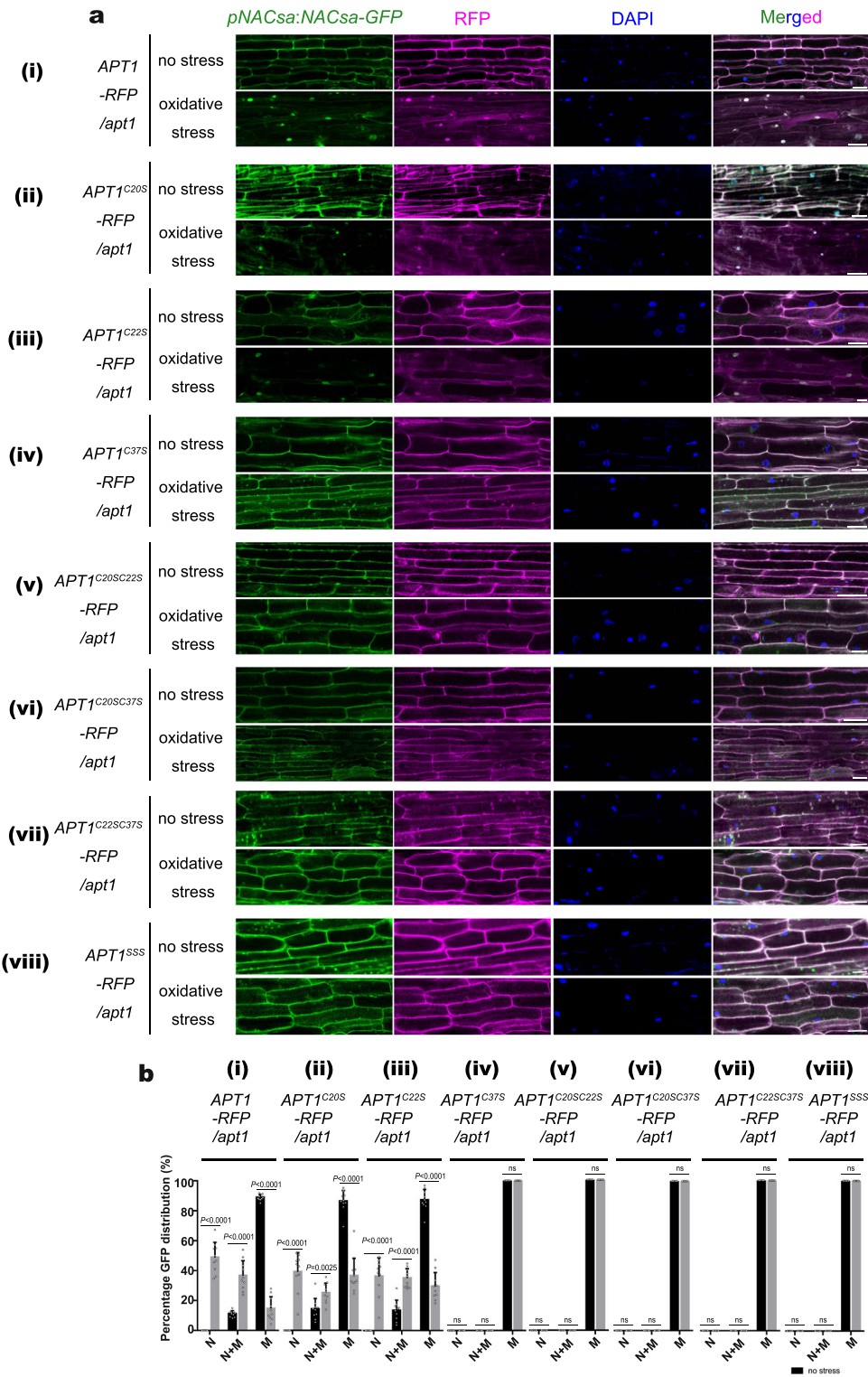

**Fig. 4 | APT1 C20/C22/C37 is oxidized by ROS to execute APT1 thioesterase activity in vivo. a** Confocal images of APT1-RFP (**i**) and APT1-RFP fusion proteins with different cysteine site mutations (**ii** APT1[C20S], **iii** APT1[C22S], **iv** APT1[C37S], **v** APT1[C20SC22S], **vi** APT1[C20SC37], **vii** APT1[C22SC37], **viii** APT1[SSS]) driven by the constitutive APT1 promoter in NACsa-GFP/*apt1* transient transgenic plant hairy roots exposed to normal or 4 mM $H_2O_2$ stress conditions for 15 min. DAPI was used to label the nucleus and excited at a wavelength of 385 nm. Bars = 20 μm. **b** Statistical analysis of APT1-RFP (**i**) and APT1-RFP fusion proteins with different cysteine site mutations

(**ii** APT1[C20S], **iii** APT1[C22S], **iv** APT1[C37S], **v** APT1[C20SC22S], **vi** APT1[C20SC37], **vii** APT1[C22SC37], **viii** APT1[SSS]) driven by the constitutive APT1 promoter in NACsa-GFP/*apt1* transient transgenic plant hairy roots exposed to normal or 4 mM $H_2O_2$ stress conditions for 15 min (**a**), which shows the percentages of cells expressing GFP at the nucleus (N), nucleus and membranes (N + M), or membranes (M) in transgenic hairy root cells. The data are presented as the means and SEs of statistical data of ten independent confocal images from **a**. *P*-values were two-sided Student's *t*-test. The source data are provided as a Source Data file.

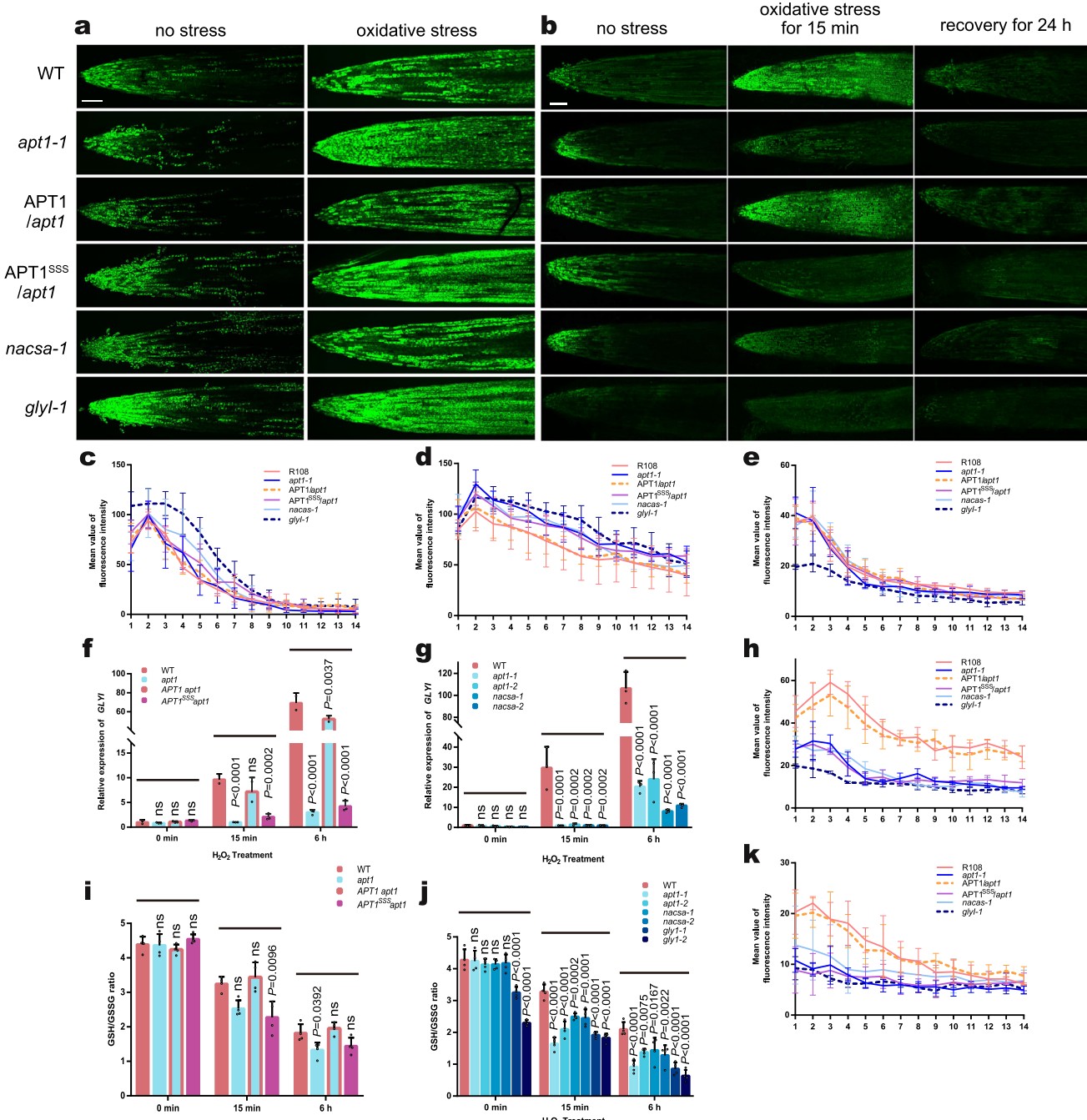

**Fig. 5 | Multimerized APT1 transports signals through NACsa and increases GSH levels to reduce ROS accumulation.** $H_2$-DCFDA (ROS) (**a**) and ThiolTracker Violet (GSH) (**b**) labelling of WT, *apt1-1* mutant, *pAPT1:APT1-3 × flag/apt1* and *pAPT1:APT1^{SSS}–3 × flag/apt1* and *nacsa, glyI* after 0 or 15 min of 4 mM $H_2O_2$ stress or during recovery after oxidative stress (15 min of 4 mM $H_2O_2$ stress followed by 24 h of nonstressed conditions in normal hydroponic solution). Bars = 100 μm. The intensity of the fluorescence in root tips was detected using a Nikon A1 laser-scanning confocal microscope (excitation at 488 nm for $H_2$-DCFDA and excitation at 405 nm for ThiolTracker Violet; emission at 525 nm for both) with the same scanning parameter settings. At least 6 roots were visualized and analysed. The mean fluorescence intensity was obtained by dividing the root length by fourteen

equal lengths using ImageJ software, and the results are shown in **c** ($H_2$-DCFDA, no stress), **d** ($H_2$-DCFDA, oxidative stress), **e** (ThiolTracker Violet, no stress), **h** (ThiolTracker Violet, oxidative stress for 15 min), and **k** (ThiolTracker Violet, recovery after oxidative stress). Relative *GLYI* relative expression and GSH/GSSG ratio in WT, *apt1-1* mutant, *pAPT1:APT1-3 × flag/apt1* and *pAPT1:APT1^{SSS}–3 × flag/apt1* (**f**, **i**) and *apt1*, *nacsa*, and *glyI* lines (**g**, **j**) after 0 h, 15 min, and 6 h of exposure to 4 mM $H_2O_2$ stress. The samples comprised the whole root of the seedling. The means and SEs were calculated from different independent replicates (**c**, **d**, **e**, **h**, **k**, *n* = 6; **f**, **g** *n* = 3; **i**, **j**, *n* = 4). *P*-values were two-sided nonparametric one-way ANOVA and Tukey's multiple range test, which compared to WT. ns means no significance with WT. The source data are provided as a Source Data file.

mutant *apt1* lines were inoculated with the pathogen causing *Medicago* root rot, namely, *Fusarium oxysporum*, and compared with the WT lines. All *apt1* mutants exhibited a lower infection rate than the WT line (Fig. 6a, b) because the expression of *GLYI* in *apt1* mutants was significantly lower than that in the WT line (Fig. 6c). *F. oxysporum*

inoculation treatment promoted the transformation of APT1 monomers to tetramers (Fig. 6d) and led to the relocation of NACsa (Fig. 6e), indicating that APT1 senses ROS triggered by biotic stress and transmits signals. ROS are related to broad-spectrum disease resistance[39–42]. Manipulation of the *APT1* level helps improve the resistance of plants

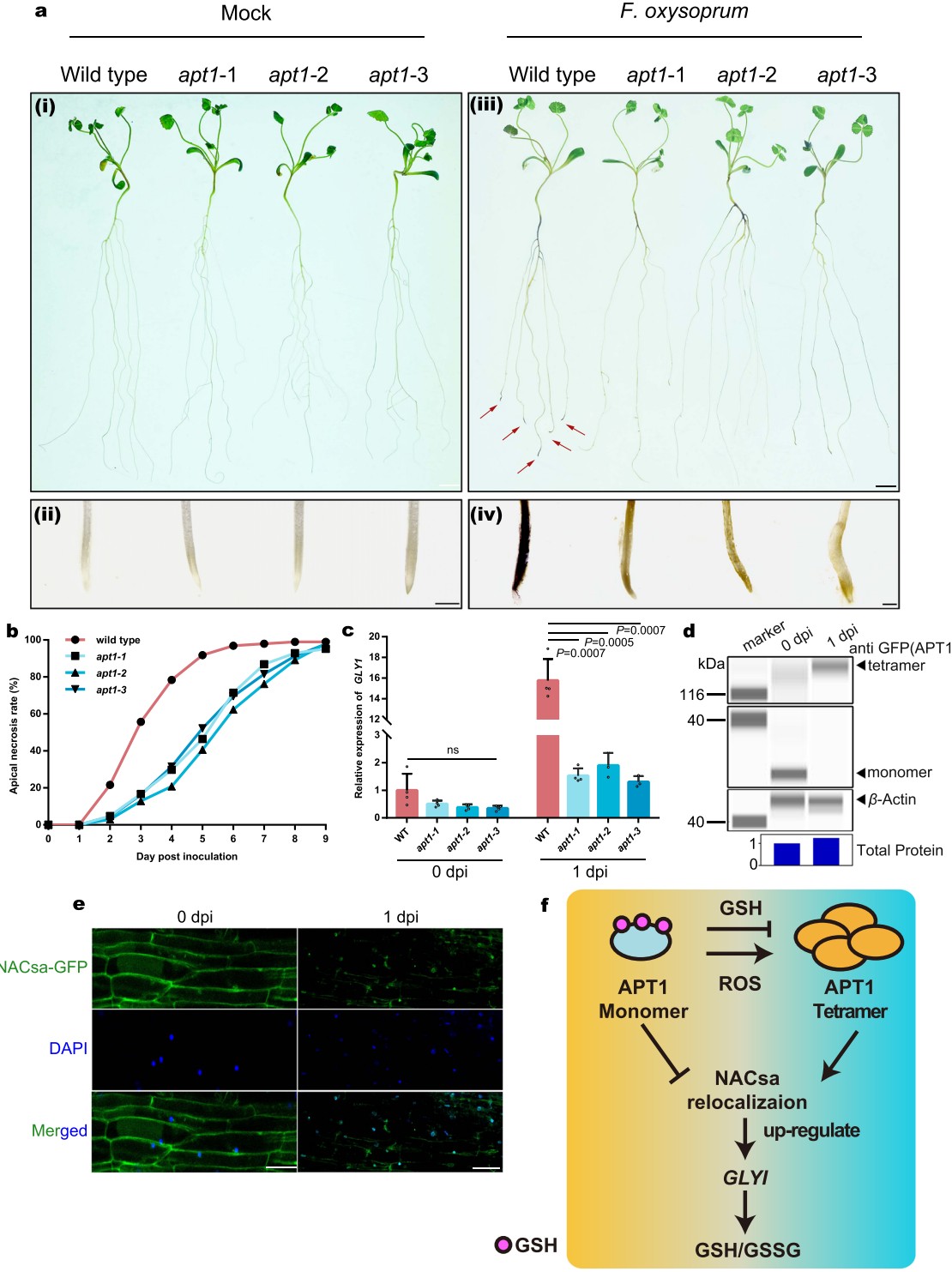

**Fig. 6 | The *apt1* mutant exhibits a higher resistance ability after inoculation with Fusarium oxysporum f. sp. medicaginis. a** Infection after inoculation with *F. oxysporum f.* sp. *medicaginis* (iii); magnified view of the root tip (iv). Control treated with water (i); magnified view of the root tip (ii). Bars = 1 cm (I and iii) or 1 mm (ii and iv). The statistics are shown in **b**, *n* ≥ 15. **c** Relative expression of *GLYI* in the WT and *apt1* mutant lines under control conditions or at 1 day post inoculation. The means and SEs were calculated from four independent replicates. *P*-values were two-sided nonparametric one-way ANOVA and Tukey's multiple range test, which compared to WT. ns means no significance with WT. **d** Simple Jess assays of transgenic *M. truncatula apt1* roots showing the multimerization of the over-expressed APT1-GFP fusion protein at 0 dpi and 1 dpi with *F. oxysporum f.* sp. *medicaginis*; detection was performed with anti-GFP (APT1) (nonreducing) (i) and anti-*β*-actin (control protein) antibodies (ii). Total protein (iii) was analysed after protein normalization for the quantification of total protein. Different parts show results from the same sample with the same loading volume (3 μl). **e** Confocal images of transgenic *M. truncatula* roots expressing NACsa-GFP fusion proteins driven by the constitutive NACsa promoter in R108 under control conditions or at 1 day post inoculation. DAPI was used to label the nucleus and was excited at a wavelength of 385 nm. Bars = 20 μm. **f** Working model of APT-mediated regulation of the redox balance. In normal states (left yellow part), APT1 is *S*-glutathionylated by GSH and forms monomers with no activity. Under oxidative stress conditions (right cyan part), ROS promote the transformation of APT1 from monomers to tetramers with enzymatic activity, which mediates NAC relocation and upregulates *GLYI* expression to increase the GSH/GSSG ratio. The source data are provided as a Source Data file.

to pathogens and thus has potential application value in plant breeding.

## Discussion

In this study, we investigated the protein properties and enzymatic activity of the previously identified depalmitoylated APT1 and found that $S$-glutathionylation of its cysteine residues inhibited its thioesterase activity. APT1 perceives the ROS signal and forms multimers in response to oxidative stress. APT1 transmits the stress signal by depalmitoylating the plasma membrane-anchored transcription factor NACsa and upregulates *GLYI* expression. APT1 activity depends on the formation of its multimers, and the activated APT-NACsa-GLYI network increases the GSH/GSSG ratio, reduces the intracellular $H_2O_2$ content and enhances resistance to oxidative stress. These results suggest that APT1 is an intracellular redox sensor that rapidly and finely tunes the redox balance in response to biotic and abiotic stress (Fig. 6f).

Regulation of the redox balance is complicated and depends on various mechanisms, including prevention, interception, and repair, to maintain the balance in plant cells. A variety of enzymatic and nonenzymatic antioxidant systems play different roles, including superoxide dismutase, peroxidase, and catalase, as well as NADPH, glutathione, ascorbic acid, tocopherols, etc. Different antioxidants play different roles at different levels, while the main role of small-molecule nonenzymatic antioxidants is the initial redox buffering capacity and the perception and transmission of oxidative signals[3,4]. Among these, cysteine/thiol is essential for plants to sense changes in the redox balance through both redox-related proteins and redox sensors[43–45]. Our study revealed that the multimerization of APT1 depends on cysteine residues and that mutations of these cysteine residues affected not only its multimerization but also the cell membrane localization of APT1 under oxidative stress conditions (Supplementary Fig. 5g, i). The cysteine-mutated APT1$^{SSS}$ does not depalmitoylate NACsa to induce its relocation to the nucleus under oxidative stress conditions, possibly because the function of thioesterase is needed for the palmitoylation of cysteines on the thioesterase to achieve membrane anchoring[32,46]. However, our results revealed that the $S$-glutathionylated APT1 monomer exhibits significantly reduced enzyme activity, which is the main factor affecting the function of the thioesterase (Figs. 3a, 4 and Supplementary Fig. 2e). GSH inhibits APT1 enzyme activity through $S$-glutathionylation, which enriches our knowledge of the effect of GSH/S-glutathionylation modification on protein function. GSH is generally known as the major thiol and has many functions[47], including stress perception, plant development, ROS removal, detoxification, protection of enzyme activity, and regulation of biological processes by $S$-glutathionization[3,23,25,38,48–52]. $S$-glutathionylation is a good characteristic of protein stability and function, and glutathione can stabilize the activity of sulfhydrylase[53]. However, with the increases in oxidative stress and oxidative damage to cells, APT1 may undergo sulfonylation after reversible disulfide bond breaks followed by irreversible degradation (Fig. 3a ii and Supplementary Fig. 5d ii).

Recent advancements show that the accumulation of ROS in the apoplast/cytoplasm and in different cell compartments under stress is different and that the working mechanism of the ROS scavenging system also varies. For example, the apoplast is more dependent on the antioxidant enzyme system, whereas in the cytoplasm, nonenzymatic small molecule antioxidants are the first to respond to oxidative stress[3,16]. ROS and other signals generated in response to different stimuli in different cell compartments can trigger stress-specific signal transduction pathways, and the mechanisms regulating the redox balance in intracellular and extracellular medium are also different[54]. Apoplastic $H_2O_2$ signal transmission is the fastest signal transduction pathway[55], but the removal of ROS is more important in crops; the intracellular sensor system plays a more direct role in ROS removal. For example, GPX1 promotes the binding of the transcription factor bZIP68 to downstream osmotic stress response genes to resist stress[56], the

response of mammalian ASK1/MTK1 to oxidative stress affects SAPK/JNK signal transduction[57], and thioredoxin, ferredoxin and other redox proteins contribute to intracellular redox homeostasis[3]. Compared with the redox sensors that sense extracellular $H_2O_2$, the intracellular redox sensors respond relatively late, but the perception of oxidative signals of the intracellular redox sensor APT1 could directly play a role through the APT1-NACsa-GLYI signalling pathway and thus regulate the redox balance fluctuations caused by changes in the ROS and GSH levels under biotic and abiotic stresses. This resistance to fluctuations can help cells cope more favourably with stress. We used various methods to measure the $H_2O_2$ level in *Medicago* after exposure to drought stress for 2 h, and all the treatments approached a concentration of 4 mM. Studies have shown that the extracellular $H_2O_2$ levels are significantly higher than the intracellular $H_2O_2$ levels; thus, the concentration of 4 mM is closer to the concentration in apoplasts. Extracellular $H_2O_2$ can enter cells through aquaporins and others[3,18], and by using fluorescence probes, we found that the intracellular $H_2O_2$ level in wild-type plants is significantly increased after exposure to oxidative stress induced by 4 mM $H_2O_2$. These results indicate that drought stress can increase the intracellular $H_2O_2$ levels, which are then sensed by APT1 and other intracellular redox sensors to initiate ROS removal.

The APT1-NACsa system regulated by ROS and GSH levels is ubiquitous in plants. APT1 is conserved in plants, and the key motif of APT1 homologous proteins in *Glycine*, *Arabidopsis* and *Solanum* show 58.67%, 54.67%, and 50.67% homology, respectively, compared with APT1 in *Medicago*. The C22 and C37 sites are highly conserved, implying that they have a common working mechanism. NACsa is also highly conserved in plants. Duan et al. showed that the membrane-anchored transcription factor NACsa relocates to the nucleus through depalmitoylation in *Medicago*, *Glycine* and *Arabidopsis*[33]. These membrane-anchored NAC transcription factors may be depalmitoylated by APT1, and different NACs may regulate different target genes after relocation to the nucleus to maintain the redox balance. The system regulating the redox balance with the redox sensor APT1 as its centre has potential breeding value, including resistance to abiotic and biotic stresses.

## Methods
### Plant materials and growth conditions
The *Medicago truncatula* ecotype R108 and the *Tnt1* insertion mutant lines NF5250 (*nacsa-1* mutant), NF9803 (*nacsa-2* mutant), NF15130 (*apt1-1* mutant), NF10049 (*glyI-1* mutant) and NF20885 (*glyI-2* mutant), which were screened from a *Tnt1* retrotransposon-tagged mutant population of *M. truncatula*, were used[58,59].

*Medicago* seedlings were grown in soil or plastic cubes containing half-strength Murashige and Skoog salts (MS) (Phyto Tech), 1.0% (w/v) sucrose (Sigma, USA), and 1.5% (w/v) agar (Sigma) or in pots filled with nutrient solution in controlled environmental rooms/plant growth chambers (Saifu, China) at 18-22 °C. During growth and treatment, the roots of the seedlings were submerged in a nutrient solution that was aerated every 2 h. The fluency rate of white light was ~100 μmol m$^{-2}$ s$^{-1}$. The photoperiod was a 16-h light/8-h dark cycle. The seeds were sown on soil or half-strength MS medium, placed in the dark at 4 °C for 3 days, and then transferred to growth rooms or chambers.

### DNA constructs and transgenic lines
Gene cloning was used to generate the *pAPT1:APT1-GFP*, *pAPT1:APT1-RFP*, *pAPT1:GUS*, *p35S:APT1-GFP*, *p35S:APT1$^{C20S}$-GFP*, *p35S:APT1$^{C22S}$-GFP*, *p35S:APT1$^{C37S}$-GFP*, *p35S:APT1$^{C20S/C22S}$-GFP*, *p35S:APT1$^{C20S/C37S}$-GFP*, *p35S:APT1$^{C22S/C37S}$-GFP*, *p35S:APT1$^{SSS}$-GFP*, and *pNACsa:NACsa-GFP* constructs. *APT1* cDNA and the *APT1* promoter region (2 kb) were amplified by PCR from *M. truncatula* ecotype A17 cDNA or genomic DNA. *apt1-2* and *apt1-3* were generated using the CRISPR/Cas9 system based on Zhu et al.[60]. The *Tnt1* mutant *apt1-1* and the knockout mutants *apt1-2* and *apt1-3* generated using CRISPR/Cas9 technology are all designated *apt1* mutants. All primers are shown in Supplementary Table 1.

The plant expression construct was transformed into the *A. tumefaciens* EHA105 strain and used for the transformation of *M. truncatula* cv. R108 as previously described[61].

## Oxidative treatment/recovery treatment of *M. truncatula* and RT–qPCR assay

*M. truncatula* seeds were germinated as described above, and the seedlings were planted in plastic pots with nutrient solution for oxidative treatment. We selected 4 mM $H_2O_2$ treatment as the oxidative stress treatment based on the study conducted by Wu et al.[7] We also detected the change in $H_2O_2$ concentration before and after drought stress (Supplementary Fig. 1a). The results showed that the $H_2O_2$ concentration in roots was $0.37 \pm 0.27$ µmol/g fresh weight under normal conditions and $5.58 \pm 3.54$ µmol/g fresh weight after 2 h of drought stress. According to a previous study[48], the corresponding value for 10 µM $H_2O_2$ in cells was 0.1 - 1 µmol/g fresh weight, whereas the ratio of the $H_2O_2$ concentration in the apoplast (extracellular) to that in the cytoplasm (intracellular) was -100:1; thus, the apoplastic $H_2O_2$ concentration corresponding to 5.58 µmol/g fresh weight is -5.58 mM. We therefore comprehensively selected treatment with 4 mM $H_2O_2$. Two-week-old seedlings were transferred to new nutrient solution with 4 mM $H_2O_2$ for different durations. The seedlings were allowed to recover after treatment with 4 mM $H_2O_2$ for 15 min or 1 h; during this recovery treatment, the roots were washed twice with water or PBS with 0.1 mM GSH to eliminate $H_2O_2$ and rinsed twice with distilled water, and the seedlings were then placed in water or nutrient solution for different times. Whole plants were utilized for the extraction of total RNA with TRIzol reagent (Invitrogen, USA), and first-strand cDNA synthesis was performed using 2.0 µg of total RNA, oligo-dT (18) and M-MLV Reverse Transcriptase (Promega, USA). Quantitative real-time RT–PCR (RT–qPCR) analysis was performed with a CFX96 real-time system (Bio-Rad, USA) using SYBR Green I (Vazyme, China). To validate the presumed stable expression of reference genes under oxidative stress, the stability of four candidate reference genes (*MtActin4A*, *MtActinE*, *MtelF4A*, and *MtGAPDH*) during oxidative stress was ranked according to a comparison of Ct values using geNorm software, which defined the internal control gene stability measure (M) as the average pairwise variation in a particular gene with all other control genes[62]. The expression of *MtActin4A* and *MtActinE* was found to be stable at the lower M value, and these were selected as the reference genes under oxidative treatment. For the temporal expression of the *MtAPT1* transcript, the relative expression data were normalized to those of *MtActin4A* and *MtActinE*. The mean values and SEs were calculated from the results of three independent experiments. The primers used in this study are listed in Supplementary Table 1.

For the spatial analysis of *MtAPT1* expression, the roots, hypocotyls, stems and leaves of WT R108 plants were sampled. For the analysis of *MtGLYI* expression after oxidation for 0 min, 15 min, and 6 h, three individual plants were sampled. The experiments were independently repeated twice. Total RNA extraction, reverse transcription and RT–qPCR analysis were performed as previously described.

## DAB or NBT staining and determination of the $H_2O_2$ level and GSH/GSSG ratio

Two-week-old hydroponic seedlings were treated with 4 mM $H_2O_2$. DAB and NBT staining were performed using the modified protocol described by Andrio et al.[63] Plant roots were incubated in 0.1 M citrate buffer (pH 3.7) supplemented with 1 mg ml$^{-1}$ DAB for 2 h. The roots were then cleared by incubation with lactic acid (10%; v/v). The plant roots were infiltrated with 10 mM sodium phosphate buffer (pH 7.8) under vacuum at room temperature for 90 min and then incubated with the staining solution (1 mM NBT, 10 mM $NaN_3$, 50 µM NADPH, and 10 mM sodium phosphate buffer, pH 7.8) for 30 min at 37 °C. The roots were then cleared twice with 80% ethanol, and all stained roots were imaged using an Olympus SZ2-ILST microscope.

The total glutathione (GSH + GSSG) and GSSG levels were measured through kinetic determination methods using a GSH and GSSG Assay Kit (Beyotime, China), and the GSH/GSSG ratio (reduced/oxidized glutathione) was calculated. To determine the total glutathione content in the samples, the plant material was collected and fully ground in a mortar with liquid nitrogen, and -0.1 g (the sample mass does not need to be precise because it does not need to be calculated in the results) of the powder was transferred to a precooled centrifuge tube. Then, 30 µL of protein removal reagent was added, the mixture was fully vortexed, and 70 µL of protein removal reagent was added. The centrifuge tube was repeatedly inverted to ensure thorough mixing, placed on ice for 10 min, and then centrifuged at $12,500 \times g$ and 4 °C for 10 min. The supernatant was collected for the determination of the total glutathione content, and the sample was diluted 10-fold. To determine the GSSG content in the sample, after centrifugation of the supernatant in the previous step, GSH was added to remove the auxiliary solution at a ratio of 20:1, and the sample was immediately vortexed and mixed thoroughly. GSH was then added at a ratio of 100:1. The working solution was removed, vortexed immediately, mixed well, and incubated at 25 °C for 1 h. The treated samples were used for determination of the GSSG content (the sample was diluted 10-fold). Subsequently, 150 µL of GSH detection working solution was added to 10 µL of sample, the mixture was incubated at 25 °C for 5 min, 50 µL of 0.16 mg/mL NADPH was added, and a Multimode microplate reader (Spark®, Tecan, Switzerland) was used to rapidly detect the total GSSG amount in the sample or standard. The levels of glutathione and GSSG were determined using a kinetic method, and these amounts were measured once at 0 min and again after 25 min. The detection wavelength was 405 nm, the vibration time was 3 s, the medium speed was medium, and the interval time was 30 s. First, ΔA405/min was calculated according to the absorbance values measured at different time points, the concentration of the standard was then taken as the abscissa, and ΔA405/min was considered the ordinate to obtain the standard curve for total glutathione or GSSG. Using the standard curve, according to the ΔA405/min value of the sample, the total glutathione or GSSG content in the sample could be calculated according to the following formula: GSH/GSSG = (total glutathione − GSSG × 2)/GSSG GSH/GSSG ratio. It should be noted that to ensure the accuracy of the data, it is necessary to perform the entire operation rapidly and on ice. The vertical bars represent the standard errors from four individual samples.

A hydrogen peroxide content detection kit (BC3590, Beijing Solarbio Science & Technology Co., Ltd, China) was used to measure the level of hydrogen peroxide in *Medicago* roots under nonstress conditions, and the samples were treated with 50% PEG-8000 to simulate drought stress (Fig. 1a i). The plant material was collected and fully ground in a mortar with liquid nitrogen. Approximately 0.1 g of the powder was transferred to a precooled centrifuge tube and reacted with titanium sulfate to generate a yellow titanium peroxide complex. After dissolving, the absorbance was measured using a spectrophotometer (UV-1900i, Shimadzu, Japan) with a wavelength of 415 nm, and the hydrogen peroxide content was calculated. We also determined the level of hydrogen peroxide by using trichloroacetic acid and CM-$H_2$DCFDA (see the Supplementary materials).

## Hairy root transformation

*pAPT1:APT1-GFP* and related cysteine mutants were transformed into WT and *apt1-1* mutants using *Agrobacterium rhizogenes*-mediated hairy root transformation as previously described[64] and subjected to different treatments, as shown in the figure legend. The transgenic plants were used for subcellular localization analysis or cellular fractionation. The immunoblot assay results were analysed by 10% SDS–PAGE and immunoblotted with a GFP monoclonal antibody (M20004M, Abmart, China, -27 kDa) at a 1:10,000 dilution for GFP tag detection and a NPTII monoclonal antibody (ab60018, Abcam, China, -29 kDa) at a 1:10,000

dilution for kanamycin resistance detection, which indicated the successful expression of the vector.

## Subcellular localization analysis

For the analysis of APT1-GFP, Cys was mutated to Ser APT1-GFP, and NACsa-GFP transgenic plants were grown in half-strength MS medium in plastic dishes for 2 weeks after the different treatments and analysed by GFP/RFP confocal imaging with a NIKON A1. Subcellular APT1-GFP in protoplasts was transiently expressed by *M. truncatula* leaves once the seedlings were 3–5 weeks of age. After the leaves were enzymatically digested, the plasmid was transferred through PEG mediation and analysed by GFP/RFP confocal imaging with a NIKON A1.

GFP fluorescence was excited at a wavelength of 488 nm, DAPI was used to label the nucleus and excited at a wavelength of 385 nm, fluorescence FM4-64 labelling was observed with an excitation wavelength of 546 nm, and single slice imaging was performed due to the thickness of the hairy root tissues.

## Protein extraction and cellular fractionation

The fusion proteins were extracted by NLB (Beyotime, China). To detect the multimers of APT1 after oxidative stress, the *p35S:APT1-GFP* stable line #25 was subjected to treatment and protein extraction for subsequent SDS–PAGE and Western blotting. To separate the soluble cytoplasm and insoluble membrane components, pAPT1:APT1-GFP fusion proteins transiently expressed in *Medicago* hairy roots treated with 4 mM $H_2O_2$ or under normal conditions were extracted and fractionated following the procedure described by Lei et al.[65] with some modifications. To isolate nuclear fraction, the abovementioned fusion proteins were extracted following the procedure described by Du et al.[66] with some modifications. In detail, the root samples were homogenized in buffer I (50 mM HEPES-KOH, pH 7.5, 10% sucrose, 50 mM NaCl, 5 mM $MgCl_2$, 1 mM 2-mercaptoethanol, 1× protease inhibitor cocktail, and 2 mM phenylmethanesulfonyl fluoride). The homogenate was centrifuged at 25,000 × g and 4 °C for 10 min, and the supernatant and precipitate were collected separately. The supernatant was centrifuged again at 100,000 × g and 4 °C for 1 h, and the supernatant sample was collected as a soluble cytoplasmic fraction. The collected sample was centrifuged in Buffer II (50 mM HEPES-KOH, pH 7.5, 10% sucrose, 50 mM NaCl, 5 mM $MgCl_2$, 1 mM 2-mercaptoethanol, 0.2% Triton X-100, 1× plant protease inhibitor mixture and 2 mM benzenemethanesulfonyl fluoride) at 25,000 × g and 4 °C for 10 min, and the supernatant was collected as the insoluble membrane fraction. To separate the cytoplasm from the nucleus, 1.5 g of root sample was homogenized with 3 mL of lysis buffer (20 mM Tris-HCl, pH 7.4, 20 mM KCl, 2 mM EDTA, 2.5 mM $MgCl_2$, 25% glycerol, 5 mM DTT, and 1× plant protease inhibitor mixture), and the supernatant was recentrifuged and collected as the cytoplasmic fraction, with 5 mL of precooled NRBT buffer (20 mM Tris-HCl, pH 7.4, 2.5 mM $MgCl_2$, 25% glycerol, and 0.2% Triton X-100) used to gently wash the precipitate three times. The sample was then resuspended in 300 µL of Extraction Buffer II (10 mM Tris-HCl, pH 8.0, 250 mM sucrose, 10 mM $MgCl_2$, 5 mM β-mercaptoethanol, 1% Triton X-100 and 1× plant protease inhibitor mixture) and gently placed in 300 µL of Extraction Buffer III (10 mM Tris-HCl, pH 8.0, 1.7 M sucrose, 2 mM $MgCl_2$, 5 mM β-mercaptoethanol, 0.15% Triton X-100 and 1× protease inhibitor mixture), and the precipitate was resuspended in 100 µL of lysis buffer and used as the nuclear fraction. The protein concentrations were measured using a Coomassie (Bradford) Protein Assay Kit (Thermo, USA). The soluble cytoplasm fractions, insoluble membrane fraction and nuclear fraction were adjusted to obtain equal concentrations, and the fractions (1 mg) were analysed by 10% SDS–PAGE and immunoblotted using a GFP monoclonal antibody (M20004 M, Abmart, China, ~27 kDa) at 1:10,000 dilution for GFP tag detection, $H^+$-ATPase (Agrisera, AS07260, Sweden, ~95 kDa) at 1:10,000 dilution, cFBPase (Agrisera, AS04043, Sweden, ~37 kDa) at 1:10,000 dilution and Histone H3

(Agrisera, AS10710, Sweden, ~17 kDa) at 1:10,000 dilution, which were used as markers for APT1, the plasma membrane, the cytosolic fraction and the nucleus, respectively.

## Immunoblot assays and simple Jess capillary-based electrophoresis immunoblot assays

The nonreducing protein samples were detected using a Tris-Tricine SDS–PAGE system (P1320, T1210, and T1220, Beijing Solarbio Science & Technology Co., Ltd., China), and the reducing samples were detected using a Tris-Glycine SDS–PAGE system.

Capillary-based electrophoresis and protein normalization were used to assay the multimerized form of APT1 in *Medicago*. The methods were adopted according to the manufacturer's protocol (AM-PN01, Protein Simple, USA). Protein concentrations were measured with a spectrophotometer (UV-1900i, Shimadzu, Japan) and an Enhanced BCA Protein Assay Kit (P0010, Beyotime, China). The protein concentration in the lysate was further diluted to 1.2 µg/µl (for GFP/APT1 detection) or 0.5 µg/µl (for β-actin detection) with 0.1× sample buffer. Three microlitres of the mixture was loaded into the Jess setup (Protein Simple, USA) for simple Western blot assays and protein normalization assays. Separation capillaries of 12–230 kDa were used in this experiment. For the detection of the APT1 multimer, all buffers were DTT-free, and samples were not heated. The antibody dilutions were GFP (1:50) and β-actin (1:50).

## Molecular docking

Covalent docking was conducted in MOE v2018.01[67]. 2D structures of GSH were prepared with a molecule and protein build module in MOE and converted to a 3D structure through energy minimization. The reactive covalent sites were Cys20, Cys22, and Cys37. The covalent reaction was defined in the Marvin Sketch, the side chain sulfur atom was the nucleophilic reactive site, and the thiol in GSH was the electrophilic warhead. Prior to docking, the force field of AMBER10:EHT and the implicit solvation model of the reaction field (R-field) were selected. The docking workflow followed the "induced fit" protocol, in which the side chains of the receptor binding site could move according to the ligand conformations with a constraint on their positions. The weight used for tethering side chain atoms relative to their original positions was 10. For the ligand, all docked poses were first ranked by London dG scoring, and force field refinement was then performed for the top 30 poses, followed by a rescoring of GBVI/WSA dG.

## Recombinant protein purification

The pET-30a(+)-His-APT1 and pET-30a(+)-His-APT1$^{C20S}$, pET-30a(+)-His-APT1$^{C22S}$, pET-30a(+)-His-APT1$^{C37S}$, pET-30a(+)-His-APT1$^{C20SC22S}$, pET-30a(+)-His-APT1$^{C20SC37S}$, pET-30a(+)-His-APT1$^{C22SC37S}$, and pET-30a(+)-His-APT1$^{SSS}$ constructs were transformed into *E. coli* strain BL21 (DE3), and the proteins were purified via Ni affinity chromatography according to the manufacturer's instructions.

## Liquid chromatography–mass spectrometry analysis and Western blot detection of APT1 S-glutathionylation

His-APT1 proteins were prepared as previously described and analysed by liquid chromatography–mass spectrometry. The proteins and treated samples were subjected to nonreducing treatment and sequentially digested with trypsin and GluC. Liquid-phase analysis was performed with Eksigent NanoLC 425 (SCIEX, China). The trap column was as follows: Nano cHiPLC Trap column 200 µm × 0.5 mm ChromXP C18-CL 3 µm 120 Å. The analytical column was as follows: Nano cHiPLC column 75 µm × 15 cm ChromXP C18-CL 3 µm 120 Å. Mobile phase A consisted of 98% water + 2% acetonitrile + 0.1% formic acid, and mobile phase B was 98% acetonitrile + 2% water + 0.1% formic acid. The flow rate was 300 nL/min. Mass spectrometry was performed with a TripleTOF™5600+ (SCIEX, China) instrument and the following

parameters: TOF MS, m/z 50–1500, accumulation time of 0.25 s; TOF MS was followed by 30 product ion scans with an accumulation time of 100 milliseconds per MS/MS; m/z 100-1500; dynamic exclusion times of 8 s for the 30-min gradient and 12 s for the 60-min gradient; ion spray voltage, 2.4 kV; GS1, 5; curtain gas, 30; DP, 100; rolling CE enabled; and CES, 5. The mass spectra were processed, and peptide identification was performed using MultiQuant™ (SCIEX, China). The search was performed against the APT1 sequence. A false discovery rate of 1% was assigned to the protein and peptide spectrum matches.

To detect the S-glutathionylation of APT1, Western blotting was performed with GSH monoclonal antibody (MA1-7620, Invitrogen, USA) at a 1:1000 dilution, and β-actin antibody (P60709, Abmart, China, ~42 kDa) was used as an internal control protein. Seedlings were also treated with a biotinylated analogue of GSH, BioGEE (G36000, Invitrogen, USA). Unlike GSH, BioGEE can be transferred to the cell membrane and enter the cytoplasm. For the treatment, two-week-old hydroponic seedlings were soaked in 0.1 mM BioGEE for 1 h and then treated with 4 mM $H_2O_2$. Samples were collected at different times, and Western blot detection was performed after protein extraction using streptavidin-horseradish peroxidase (HRP)-conjugated antibody (SA10001, Invitrogen, USA) at a 1:5000 dilution. We tested the specificity of the antibodies, including anti-GFP, anti-GSH and anti-biotin, and observed high specificity (Supplementary Fig. 5a–c).

### MST analysis
MST analysis was performed using a NanoTemper Monolith NT.LabelFree instrument (NanoTemper Technologies GmbH, Germany). For each assay, the APT1 protein (50 nM) was mixed with an equal volume of ligand (GSH or Cys) at 16 different serial concentrations in buffer (10 mM PBS, pH 7.4, and 137 mM NaCl) at room temperature. The samples were loaded into standard glass capillaries (Monolith NT.LabelFree capillaries), and thermophoresis analysis was performed (LED 40%, medium MST power) with MO. Control software (NanoTemper Technologies GmbH, Germany). For each set of binding experiments, three independent MST measurements were obtained at 360 nm. The datasets were processed with MO.Affinity Analysis software (NanoTemper Technologies GmbH, Germany).

### Histochemical GUS activity analysis
Histochemical staining for GUS activity was performed using APT1 promoter-driven GUS (pAPT1:GUS) transgenic lines. Seedlings grown in half-strength MS medium or soil were used for histochemical staining. Data representing three independent lines, #2, #11, and #13, were examined, and all three lines displayed similar staining patterns.

### Pathogen preparation and inoculations
*Fusarium oxysporum* was grown on agar plates. The seedlings were grown in half-strength MS medium in plastic cubes, and when the seedlings were 2 weeks of age, they were treated with *F. oxysporum* spore solution ($10^6$ spores ml$^{-1}$). The rotten root tips in >15 plants of each line were counted every day starting from the first day of the treatment. *GLYI* expression and NACsa-GFP subcellular localization were analysed as previously described.

### Fluorescent probe detection for visualizing ROS and GSH production in root tips
ROS production in root tips was visualized via the fluorescence probe 2,7-dichlorodihydrofluorescein diacetate acetyl ester (H₂DCFDA) (D6883, Sigma, USA)[68]. Two-week-old hydroponic seedlings of WT (cv. R108) and transgenic lines were transferred into water or treated with 4 mM $H_2O_2$ for 15 min. The treated plant roots were then washed five times with 10 mM MES-KCl buffer (pH 6.1). The plant roots were immersed in 10 mM MES-KCl buffer with 50 μM H₂DCFDA solution. After incubation for 15 min in darkness, the roots were washed five times with deionized water. The fluorescence intensity in root tips was

detected using a Nikon A1 laser-scanning confocal microscope (excitation 488 nm, emission 525 nm) with the same scan parameter settings: laser HV(GaAsP) of 50, laser value of 2.0, and pinhole value of 1.2. At least 6 roots were visualized and analysed. The mean fluorescence intensity was obtained by dividing the root length by fourteen equal lengths and analysed using ImageJ software. Negative controls without fluorescent probes and analysed using the same detection conditions are shown in Supplementary Fig. 5h. The significance of the mean value of ROS detection fluorescence intensity of each point of Fig. 5c, d are shown in Supplementary Table 2.

The production of GSH in root tips was visualized via the fluorescence probe ThiolTracker™ Violet dye (T10095, Invitrogen, USA)[69]. Two-week-old hydroponic seedlings of WT (cv. R108), and transgenic lines were transferred into water or treated with 4 mM $H_2O_2$ for 15 min and allowed to recover in water after 4 mM $H_2O_2$ treatment for 15 min. The treated plant roots were then washed twice with 10 mM Dulbecco's PBS (pH 7.2) and immersed in 10 mM Dulbecco's PBS with 20 μM ThiolTracker™ Violet dye solution. After incubation for 30 min (RT) in darkness, the roots were washed three times with Dulbecco's PBS. The fluorescence intensity in root tips was detected using a Nikon A1 laser-scanning confocal microscope (excitation at 405 nm, emission at 526 nm) with the same scan parameter settings: laser HV(GaAsP) of 100, laser value of 2.0, and pinhole value of 1.2. At least 6 roots were visualized and analysed. The mean fluorescence intensity was determined by dividing the root length by fourteen equal lengths and analysing the images with ImageJ software. The significance of the mean value of the GSH detection fluorescence intensity of each point of Fig. 4e, h, and k is shown in Supplementary Table 3.

### Assay of APT1 activity
APT1 was purified as described above. Enzyme activity was measured using a Multimode microplate reader (Spark®, Tecan, Switzerland) based on the release of 4-nitrophenolate from 4-nitrophenyl caprylate[70]. The assay was performed in buffer containing 20 mM HEPES and 150 mM NaCl titrated to pH 7.4 using 2 N NaOH solution. In this assay, 80 μL of a solution consisting of 30 μg of APT1 protein in buffer containing 0.01% (v/v) Triton-X was mixed. Subsequently, 20 μL of an emulsion of 5 mM 4-nitrophenyl caprylate in 20 μL of 0.25% (v/v) Triton-X in HEPES buffer was added to the reaction mixture. The absorbance was measured at 401 nm, and the data were recorded from 0 min to 300 min. During the 300-min measurement period, the reaction mixture was shaken at 540 rpm for 10 s at 50-s intervals. Linear regression was performed using GraphPad Prism 7 software. The data were assessed based on an enzymatic reaction progress curve (Supplementary Fig. 2e).

### Statistical analysis
The values are presented as the means ± SEs. Assessments of significance were performed by nonparametric one-way ANOVA and Tukey's test using SPSS statistical software. The P-values from each statistical test are reported.

### Reporting summary
Further information on research design is available in the Nature Portfolio Reporting Summary linked to this article.

## Data availability
Source data are provided with this paper.

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

## Acknowledgements

This work was supported by the Science and Technology Innovation 2030-Major Project (2022ZD04012, J.D.), the Key Projects in Science and Technology of Inner Mongolia (2021ZD0031, J.D.) and the National Natural Science Foundation of China (32070272, T.W.). We thank Yinghua Zhao (Shanghai AB Sciex Analytical Instrument Trading Co.) for helping with the LC-MS/MS analyses; Qing Chang, Yalan Chen and Zi Yang (School of Life Science, Tsinghua University) for helping with the MST analyses; Yanli Zhang (School of Life Science, Tsinghua University) for helping with the subcellular localization analysis; Mei Song (Beijing TOPBIOX Technology Co., Ltd.) for helping with the simple Jess capillary-based electrophoresis assays; and Qijun Chen (College of Biological Science, China Agriculture University) for providing the CRISPR/Cas9 toolkit.

## Author contributions

T.W. and J.D. directed the research. T.J. performed the main experiments and data analyses. L.Z., J.Wu and M.D. contributed to the transformation work. Q.L., P.L., C.S., J.L., and Q.Y. contributed to the molecular cloning work. J. Wen provided the *Tnt1* mutant materials. T.W. and J.D. oversaw the entire study. T.J., T.W., and J.D. wrote the manuscript.

## Competing interests

The authors declare no competing interests.
