## [Peer Review File · Nature Communications]

The thioesterase APT1 is a bidirectional-adjustment redox sensorEditorial Note: This manuscript has been previously reviewed at another journal that is not operating a transparent peer review scheme. This document only contains reviewer comments and rebuttal letters for versions considered at *Nature Communications*.

REVIEWER COMMENTS

Reviewer #1 (Remarks to the Author):

The authors have performed significant amount of experiments and the revision has answered the majority of my prior concerns. The use of thiol tracker to show GSH accumulation in different genetic materials and measurement of APT1 enzymatic activity in vitro are convincing. However, the claim that APT1 is a sensor for both ROS and GSH is too strong. While this may be the case for GSH, it is not known what concentration of H₂O₂ APT1 responds to. It is necessary to tone down the claim. Ideally, the authors need to use the reduced form of recombinant APT1 and react with different concentrations of H₂O₂ to determine the the concentration that oxidizes APT1. Exogenously applied H₂O₂ to hairy roots causes oligomerization and PM localization of APT1, this is not direct evidence of APT1 sensing H₂O₂.

Other comments:

1. How was H₂O₂ measured in Fig. S1A? I looked through the entire manuscript and methods, but did not see any ddescription.
2. Fig. 2D, decreasing concentrations of APT1 were tested. But the immunoblot seems to show the same amount of protein. Did the author used decreasing amount of protein in incubation but adjusted amount of protein before loading the gel? This should be described more clearly.
3. Throughout of the text (such as line 118), "nonreduced SDS-PAGE" should be "nonreducing".
4. Lines 146-147 "we used site-directed mutagenesis to purify APT1 protein with different cysteine mutation forms...". This needs to be rephrased.
5. Lines 148-149, "The immunoblot results showed that only APT1 with all three cysteine mutated could not be oligomerized". This statement is not accurate, as C37S, C20/22S, C20/37S, C22/37S also showed less oligomerization.

Reviewer #2 (Remarks to the Author):

The manuscript by Ji et al. is the revised version of an earlier manuscript. The work shows that the thioesterase APT1 is maintained inactive by glutathionylation but is activated and multimerizes upon shifts in the redox balance of the cell or aiotic/abiotic stress. The authors have significantly improved the manuscript with their revision and have addressed many of my concerns. However, I have few more questions/suggestions that I would like the authors to address.

1. The authors explain that the cysteines should not participate in the catalytic core of APT1. However, I am wondering whether the APT1SSS variant still displays thioesterase activity in vitro at least. Does APT1m still localize to the membrane?

2. Protein amounts in Western blots are still very uneven, for example Fig. 3A. and the GSP-APT1 Western blots are very blurry. The fact that beta-actin is degraded (or at least abundance is reduced from 12 hours onwards) makes it a poor control for the Western blot experiments. Are the cells still viable at that timepoint? If not, the results at those late timepoints are no longer biologically meaningful. If the cells are viable, another stably expressed protein should be used as a loading control. It is also puzzling to see such a long-term effect of 4 mM H₂O₂, since likely there will be only extremely little H₂O₂ left after few hours of treatment.

3. On several occasions the figure panels are not referenced in order in the text.

Reviewer #3 (Remarks to the Author):

Many of the concerns regarding the earlier version of this manuscript have been addressed in revision. The authors provide convincing evidence that acyl-protein thioesterase 1 (APT1) is regulated by protein glutathionylation. Data are presented showing that the S-glutathionylated form of APT1 is a monomer without any thioesterase activity. In contrast, the oxidised tetrameric form of APT1 hydrolyses the thioester bond of the NACs transcription factor resulting in its release from the cytoplasmic membrane and relocation into the nucleus. While these data are convincing, I find some of the conclusions are too broad and not supported by data. For example, statements such as those in lines 67/68 that “The resulting higher levels of GSH removed excessive ROS and provided resistance to oxidative stress”, are likely to be incorrect because GSH is only part of a much wider antioxidant system that functions to process ROS. Moreover, S-glutathionylation is mediated by thiol-disulfide exchange with oxidized glutathione, a reaction with oxidant-induced protein thiyl radicals with reduced glutathione, or reaction of a nitrosothiol with another thiol. Thus, the explanation in lines 69-70 that “during recovery, reduced

ROS levels and high levels of GSH promoted APT1 monomerization and S-glutathionylation”, is also inaccurate. Similarly, lines 260-262 state that “APT1acts as a cellular detoxifier and a factor inducing elevations in the GSH/GSSG levels after a ROS burst”. This is something of an overstatement because many stress-induced transcription factors are able to generally enhance the expression of antioxidant enzymes. This is probably also true of NACsa but the authors have only focussed on the glutathione system in the present study and so effects on other antioxidant systems have not been investigated. The focus on the glutathione system is therefore somewhat misleading.

Lines 284-287 are also somewhat misleading. Glutathionylation is a well characterised phenomenon that has diverse effects on protein stability and functions. To state that “our research reveals a new function of GSH”, is overstating the importance of the finding. Similarly, I am not convinced that the findings have any relationship to apoplastic hydrogen peroxide signal transmission. As stated above, APT1 is not unique in triggering processes that activate antioxidant systems in response to stress. It is important that the authors acknowledge this point and discuss the role of APT1 in the broader context of overall plant responses to abiotic stress

REVIEWER COMMENTS

Reviewer #1 (Remarks to the Author):

The authors have performed significant amount of experiments and the revision has answered the majority of my prior concerns. The use of thiol tracker to show GSH accumulation in different genetic materials and measurement of APT1 enzymatic activity in vitro are convincing. However, the claim that APT1 is a sensor for both ROS and GSH is too strong. While this may be the case for GSH, it is not known what concentration of H₂O₂ APT1 responds to. It is necessary to tone down the claim. Ideally, the authors need to use the reduced form of recombinant APT1 and react with different concentrations of H₂O₂ to determine the the concentration that oxidizes APT1. Exogenously applied H₂O₂ to hairy roots causes oligomerization and PM localization of APT1, this is not direct evidence of APT1 sensing H₂O₂.

1.-Thank you for your comment. We examined the effect of using recombinant APT1 exposed to different H₂O₂ concentrations. The APT1 multimers were degraded and the level of monomers increased after application of 10 mM GSH. When applying 10 mM GSH and then adding different concentrations of H₂O₂ (Fig. 2b and Supplementary Fig. 2d), the level of APT1 monomers decreased, and multimer levels increased. We added the data in lines 119-122, Figure 2b and Supplementary Figure 2d in the revised manuscript. Thank you again for your comments, which have greatly increased the credibility of APT1 as a redox sensor.

Other comments:

1. How was H₂O₂ measured in Fig. S1A? I looked through the entire manuscript and methods, but did not see any ddescription.

--Thank you for your comment. The method of H₂O₂ detection is described below.

" A hydrogen peroxide content detection kit (BC3590, Beijing Solarbio Science & Technology Co., Ltd, China) was used to detect the level of hydrogen peroxide in *Medicago* roots under nonstressed conditions and treated with 50% PEG-8000 to simulate drought stress. The plant material was collected and fully ground in a mortar with liquid nitrogen. Approximately 0.1 g of the powder was transferred to a precooled centrifuge tube and reacted with titanium sulfate to generate a yellow titanium peroxide complex. After dissolving, the absorbance was detected using a spectrophotometer (UV-1900i, Shimadzu, Japan) with a wavelength of 415 nm, and the hydrogen peroxide content was calculated."

We have described this information in the "DAB or NBT staining and determination of the H₂O₂ level and GSH/GSSG ratio" section of the Materials and Methods in lines 429-437 of the revised manuscript.

2.Fig. 2D, decreasing concentrations of APT1 were tested. But the immunoblot seems to show the same amount of protein. Did the author used decreasing amount of protein in incubation but adjusted amount of protein before loading the gel? This should be

described more clearly.

-Thank you for your comment. We used the same amount of the APT1 protein in different volumes of a 10 mM GSH mixture when loading the gel. As you suggested, a clear description is important, and thus we clearly described the different states of APT1 observed after different treatments, including the amount, concentration and other characteristics, in lines 849-853 of the revised manuscript.

3. Throughout of the text (such as line 118), "nonreduced SDS-PAGE" should be "nonreducing".

-Thank you for your correction. All instances of "nonreduced SDS-PAGE" have been corrected to "nonreducing SDS-PAGE".

4. Lines 146-147 "we used site-directed mutagenesis to purify APT1 protein with different cysteine mutation forms...". This needs to be rephrased.

-Thank you for your comment. We rephrased the statement in lines 142-145 of the revised manuscript as follows:

"Furthermore, we purified APT1 proteins with different site-directed mutation forms, including APT1^{C20S}, APT1^{C22S}, APT1^{C37S}, APT1^{C20SC22S}, APT1^{C20SC37S}, APT1^{C22SC37S} and APT1^{C20SC22SC37S} (APT^{SSS}), then detected the proteins by nonreducing SDS-PAGE."

5. Lines 148-149, "The immunoblot results showed that only APT1 with all three cysteine mutated could not be oligomerized". This statement is not accurate, as C37S, C20/22S, C20/37S, C22/37S also showed less oligomerization.

-Thank you for your correction. We have rewritten the statement based on your comments, adding the following text: " The immunoblot results showed that APT1, APT1^{C20S} and APT1^{C22S} were multimerized; APT1^{C37S}, APT1^{C20SC22S}, APT1^{C20SC37S}, APT1^{C22SC37S} showed less multimerization; APT1 with all three cysteines mutated (APT^{SSS}) was not multimerized " in lines 145-148 of the revised manuscript.

Reviewer #2 (Remarks to the Author):

The manuscript by Ji et al. is the revised version of an earlier manuscript. The work shows that the thioesterase APT1 is maintained inactive by glutathionylation but is activated and multimerizes upon shifts in the redox balance of the cell or aiotic/abiotic stress. The authors have significantly improved the manuscript with their revision and have addressed many of my concerns. However, I have few more questions/suggestions that I would like the authors to address.

-Thank you for your positive comments on the last version of the manuscript. We have added much more data based on your previous comments, which has greatly improved the quality of our study. We have also performed additional experiments that are included in this version based on your comments, which are listed below.

1. The authors explain that the cysteines should not participate in the catalytic core of - Thank you for your comments. We further tested the enzymatic activity of APT1^{SSS}. Notably, according to a previous study (Dekker, et al. 2010), 4-nitrophenyl caprylate is the substrate used to detect esterase hydrolysis activity, and not a specific substrate of thioesterase. Our results show that APT1^{SSS} possesses hydrolytic activity compared with APT1+10 mM GSH treatment, which is shown in Fig. 1. Therefore, the catalytic core of APT1 does not contain a cysteine. However, we are more concerned that under physiological conditions *in vivo*, APT1^{SSS} does not hydrolyse thioester bonds or promote the depalmitoylation of NACsa.

Figure 1. Curve showing the progress of the enzymatic reaction for APT1, APT1+10 mM GSH, APT1^{SSS} within 300 min. The results are presented as the concentration of the product 4-nitrophenolate. n=3.

Additionally, APT1m can localize to the cytoplasmic membrane and cytoplasm. Our previous research (Duan et al., 2017) examined the coexpression of NACsa and APT1m in onion epidermal cells and suggested that NACsa and APT1m colocalize on the cytoplasmic membrane (Fig. 2 ii) but do not promote the nuclear relocalization of NACsa like APT1.

Figure 2. Co-transformation of MfNACsa-eGFP with MtAPT1-RFP (i) or MtAPT1m-RFP (DXXGXV-AXXAXA) mutant protein (ii) in onion epidermal cells. The signal

patterns were observed at 488 nm (eGFP), 546 nm (RFP) and 358 nm (DAPI) fluorescence excitation wavelength by confocal imaging (Olympus FluoView™ FV1000). Bars = 50 μ m.

2. Protein amounts in Western blots are still very uneven, for example Fig. 3A. and the GSP-APT1 Western blots are very blurry. The fact that beta-actin is degraded (or at least abundance is reduced from 12 hours onwards) makes it a poor control for the Western blot experiments. Are the cells still viable at that timepoint? If not, the results at those late timepoints are no longer biologically meaningful. If the cells are viable, another stably expressed protein should be used as a loading control. It is also puzzling to see such a long-term effect of 4 mM H₂O₂, since likely there will be only extremely little H₂O₂ left after few hours of treatment.

-Thank you for your comment. After 12 hours, the abundance of β -actin was reduced, which may indicate that no biological significance would be observed after this time point. Therefore, based on your comments and because the Western blots were uneven, we repeated this experiment using simple Jess capillary-based electrophoresis (supported by ProteinSimple, the method was described in lines 507-523 for the revised manuscript, references for simple Wes/Jess include studies by Alterman et al., 2019; Li et al., 2017; Yang et al., 2019) to detect the multimerization of APT at key time points, including 0 min, 15 min, 1 h and 6 h of oxidative treatment. The simple Jess capillary-based electrophoresis assays also produced the same result; namely, APT1 is a monomer under no stress treatment and forms a multimer after oxidative stress. We retained β -actin as a housekeeping gene and performed total protein normalization to show the consistency of the loaded protein sample. The results are shown in Fig. 3a and 3b, Fig. 5d, and lines 153-169 in the revised manuscript. In addition, we retained the WB results from nonreducing SDS-PAGE gels, which are shown in Supplementary Fig. 5.

3. On several occasions the figure panels are not referenced in order in the text.

-Thank you for your comments. We have tried our best to correct the order of all the figure panels, including Fig. 1, Fig. 2, Fig. 5 and the supplementary information, but the order of Fig. 1c and Fig. 1d is difficult to arrange due to the layout of the figures; we hope you can understand.

References

- Alterman, J.F., Godinho, B.M.D.C., Hassler, M.R. et al. A divalent siRNA chemical scaffold for potent and sustained modulation of gene expression throughout the central nervous system. *Nat Biotechnol* 37, 884–894. (2019)
- Dekker, F.J., et al. Small-molecule inhibition of APT1 affects Ras localization and signaling. *Nat. Chem. Biol.* 6, 449-456. (2010).
- Duan, M. et al. A Lipid-Anchored NAC Transcription Factor Is Translocated into the Nucleus and Activates Glyoxalase I Expression during Drought Stress. *Plant Cell* 29, 1748-1772, (2017).
- Li, Q., Lowey, B., Sodroski, C. et al. Cellular microRNA networks regulate host

dependency of hepatitis C virus infection. *Nat Commun* 8, 1789 (2017)
Yang, Y., Willis, T.L., Button, R.W. et al. Cytoplasmic DAXX drives SQSTM1/p62 phase condensation to activate Nrf2-mediated stress response. *Nat Commun* 10, 3759 (2019)

Reviewer #3 (Remarks to the Author):

Many of the concerns regarding the earlier version of this manuscript have been addressed in revision. The authors provide convincing evidence that acyl-protein thioesterase 1 (APT1) is regulated by protein glutathionylation. Data are presented showing that the S-glutathionylated form of APT1 is a monomer without any thioesterase activity. In contrast, the oxidised tetrameric form of APT1 hydrolyses the thioester bond of the NACsa transcription factor resulting in its release from the cytoplasmic membrane and relocation into the nucleus. While these data are convincing, I find some of the conclusions are too broad and not supported by data.

For example, statements such as those in lines 67/68 that “The resulting higher levels of GSH removed excessive ROS and provided resistance to oxidative stress”, are likely to be incorrect because GSH is only part of a much wider antioxidant system that functions to process ROS.

-Thank you for your comments. We agree with you that a wide range of antioxidant systems are mobilized for the process of plant antioxidant defences in response to stresses. The resistance provided by the detoxifying antioxidant system of APT1-NACsa-GLYI is part of a broader antioxidant pathway. For plants, both enzymatic and nonenzymatic antioxidant systems are potent contributors to antioxidant mechanisms, including superoxide dismutase, peroxidase, catalase, NADPH, glutathione, ascorbic acid, tocopherols, etc. Glutathione is one of these molecules and is also important for its ability to provide antioxidant activity, maintain a negative redox potential and participate in signal transduction mediated by posttranslational modifications (Hernández et al., 2015). Glutathione acts as a buffer regulator, both helping plants temporarily buffer drastic changes in oxidative stress when stress occurs to await further activation of antioxidant enzymes (Hasanuzzaman et al., 2017) and functioning to process ROS. We have changed the relevant statement to "NACsa that has relocated to the nucleus up-regulates *GLYI* expression, leading to an increase in the GSH/GSSG ratio and a decrease in the intracellular H₂O₂ content, resulting in an increased antioxidant capacity of the plant." in lines 62-65.

Moreover, S-glutathionylation is mediated by thiol-disulfide exchange with oxidized glutathione, a reaction with oxidant-induced protein thiyl radicals with reduced glutathione, or reaction of a nitrosothiol with another thiol. Thus, the explanation in lines 69-70 that “during recovery, reduced ROS levels and high levels of GSH promoted APT1 monomerization and S-glutathionylation”, is also inaccurate.

-Thank you for your comments. APT1 was present in a monomeric state at recovery for 96 h. Monomeric APT1 is likely to be the newly synthesized S-glutathionylated APT1,

whereas the multimerized APT1 is likely to have been degraded after performing its function. We have also removed the statement in lines 69-70 of the last version of the manuscript and instead elaborated on more possibilities in lines 65-66 and the Discussion section of the revised manuscript.

Similarly, lines 260-262 state that “APT1acts as a cellular detoxifier and a factor inducing elevations in the GSH/GSSG levels after a ROS burst”. This is something of an overstatement because many stress-induced transcriptions factors are able to generally enhance the expression of antioxidant enzymes. This is probably also true of NACsa but the authors have only focussed on the glutathione system in the present study and so effects on other antioxidant systems have not been investigated. The focus on the glutathione system is therefore somewhat misleading.

-Thank you for your comments. We analysed the RNA-seq data from NACsa OE transgenic materials under stress (Duan et al., 2017) (accession number PRJNA383288 in the NCBI database) as you suggested and found that NACsa did not transcriptionally regulate classical enzymatic antioxidant enzymes, such as SOD, POD, CAT, and APX, under stress conditions. However, *GLYI* expression was upregulated approximately 600-fold (Fig. 3) in response to stress, and thus we concluded that NACsa-mediated regulation of *GLYI* is the main effective pathway for the resolution of oxidative stress. We have changed this statement to "APT1 activity depends on the formation of its multimers and the activated APT-NACsa-GLYI network increases the GSH/GSSG ratio, reduces the intracellular H₂O₂ content and enhances resistance to oxidative stress." in lines 257-259 of the revised manuscript.

Figure 3 MfNACsa positively regulates the stress-, lipid transport- and localization-related genes expression under PEG-imposed drought stress. The relative expression of the candidate genes in ectopic expression of MfNACsa lines (OE23 and OE33)

compared with the WT plants under 50% PEG-8000 treatment for 4 h (A) and normal conditions (B). The gene expression was calculated using the $2^{-\Delta\Delta CT}$ method and the MtActin as an endogenous control. The mean values and SE were calculated from the three independent experiments (Kruskal-Wallis non-parametric test, *P < 0.05, **P < 0.01).

Lines 284-287 are also somewhat misleading. Glutathionylation is a well characterised phenomenon that has diverse effects on protein stability and functions. To state that “our research reveals a new function of GSH”, is overstating the importance of the finding.

-Thank you for your comments. S-glutathionylation is a well characterized indicator of protein stability and function, and glutathione stabilizes the sulfhydrylase activity and maintains the high activity state of the active centre of sulfhydrylase (Forman et al. 2009). However, in our study, S-glutathionylation inhibited the thioesterase activity of APT1. We changed the statement of the new function of GSH to "Our research revealed that GSH inhibits APT1 enzyme activity through S-glutathionylation, which enriches our knowledge of the effect of GSH/S-glutathionylation modification on protein function" in lines 284-289 of the revised manuscript.

Similarly, I am not convinced that the findings have any relationship to apoplastic hydrogen peroxide signal transmission. As stated above, APT1 is not unique in triggering processes that activate antioxidant systems in response to stress. It is important that the authors acknowledge this point and discuss the role of APT1 in the broader context of overall plant responses to abiotic stress

--Thank you for your comments. In this study, APT1 was identified as an intracellular redox sensor, and we discussed the function of the APT1-NACsa-GLYI regulatory system. We agree that the downstream transcription factor regulated by APT1 may not be unique, but Fig. 4a and 4b (in the revised manuscript) show that in the absence of stress, *glyI* mutation caused a significant increase in the H₂O₂ content in root tips (Supplementary Table 2) and a significant decrease in the GSH content (Supplementary Table 3). However, under oxidative stress conditions, no significant difference in the H₂O₂ content and GSH content was observed between *apt1* and *glyI* mutants. The APT1-NACsa-GLYI pathway is an effective system mediating the plant response to oxidative stress. As you mentioned, other redox sensors may exist in the cytoplasm that respond to oxidative stress and activate the antioxidant system, such as GPX1 in rice or MTK1 in mammals. We also discussed this possibility in lines 291-304 of the revised manuscript.

References

- Duan, M. et al. A Lipid-Anchored NAC Transcription Factor Is Translocated into the Nucleus and Activates Glyoxalase I Expression during Drought Stress. *Plant Cell* 29, 1748-1772, (2017).
- Forman, H.J., Zhang, H., Rinna, A. Glutathione: overview of its protective roles, measurement, and biosynthesis. *Mol Aspects Med.* 30(1-2):1-12. (2009).

Hasanuzzaman, M., Nahar, K., Anee, T.I., and Fujita, M. Glutathione in plants: biosynthesis and physiological role in environmental stress tolerance. *Physiology and Molecular Biology of Plants* 23, 249-268. (2017).

Hernández, L.E., Sobrino-Plata, J., Montero-Palmero, M.B., Carrasco-Gil, S., Flores-Cáceres, M.L., Ortega-Villasante, C., and Escobar, C. Contribution of glutathione to the control of cellular redox homeostasis under toxic metal and metalloids stress. *Journal of Experimental Botany* 66, 2901-2911. (2015).

REVIEWER COMMENTS

Reviewer #1 (Remarks to the Author):

I am pleased to say the authors have addressed all my concerns, and I support publication of this study.

Reviewer #2 (Remarks to the Author):

The work by Ji et al. shows that the thioesterase APT1 is maintained inactive by glutathionylation but is activated and multimerizes upon shifts in the redox balance of the cell or aiotic/abiotic stress. Ji et al. have addressed all the comments I raised towards their revised manuscript to my satisfaction. The current version of the manuscript is much improved. In my opinion, they also addressed the comments of the other reviewers adequately and the manuscript

Reviewer #3 (Remarks to the Author):

This innovative and interesting paper describes the regulation of acyl-protein thioesterase 1 (APT1) by cellular redox status, which regulates the transition between monomeric and polymeric forms. Data are presented showing that in the absence of stress, APT1 exists as a monomer through S-glutathionylation which inhibits enzymatic activity. Oxidation results in tetramerization of the protein leading to activation and translocation to the nucleus. These data are sound and convincing.

I am less convinced by the data showing that resultant increases in the cellular GSH/GSSG ratio are achieved through regulation of Glyoxalase I (GLYI), rather than posttranslational modulation of GSH1, which was not investigated. I am also concerned by the method used to measure hydrogen peroxide, which is not specific and prone to artefacts particularly when used on crude plant extracts.

However, the results concerning the mechanism of APT1 regulation are significant and important.

1 REVIEWER COMMENTS

2
3 Reviewer #3 (Remarks to the Author):

4
5 This innovative and interesting paper describes the regulation of acyl-protein thioesterase
6 1 (APT1) by cellular redox status, which regulates the transition between monomeric and
7 polymeric forms. Data are presented showing that in the absence of stress, APT1 exists
8 as a monomer through S-glutathionylation which inhibits enzymatic activity. Oxidation
9 results in tetramerization of the protein leading to activation and translocation to the
10 nucleus. These data are sound and convincing.

11
12 I am less convinced by the data showing that resultant increases in the cellular GSH/GSSG
13 ratio are achieved through regulation of Glyoxalase I (GLYI), rather than posttranslational
14 modulation modulation of GSH1, which was not investigated. I am also concerned by the
15 method used to measure hydrogen peroxide, which is not specific and prone to artefacts
16 particularly when used on crude plant extracts.

17
18 However, the results concerning the mechanism of APT1 regulation are significant and
19 important.

20
21 -Thank you for your comments. According to your opinion, we tested the H₂O₂ content
22 before and after drought stress by treatment with trichloroacetic acid (TCA)¹ and CM-
23 H₂DCFDA², as described below. The results are shown in Supplementary Fig. 1a ii and
24 iii. The results show that the H₂O₂ content before drought stress measured by TCA
25 treatment was 0.340 μmol/g fresh weight, increasing to 3.611 μmol/g fresh weight after
26 stress, which was close to the content of H₂O₂ determined by the previously described
27 method. Further detection by CM-H₂DCFDA showed that although units for the results
28 were fluorescence intensity/mg protein extract, the H₂O₂ content after treatment was
29 approximately 10 times higher than that before treatment, which was also similar to the
30 results obtained in the previous two methods (Supplementary Fig. 1 a ii and iii, in
31 revised manuscript).

Supplementary Fig. 1

a Differences in the H₂O₂ content in 2-week-old seedling roots of wild-type *Medicago* R108 before and after drought (dehydration) stress for 2 h simulated by 50% PEG-8000,

36 detected by using titanium sulfate (i), trichloroacetic acid (ii) and CM-H₂DCFDA (iii)
37 (n≥29, Tukey nonparametric test, *** indicates P<0.001).

38

39 We detected the enzyme activity of GSH1, as shown in the Supplementary Fig. 9 a. The
40 results showed that there was no significant difference between mutants and WT under
41 oxidative stress for 15 minutes. After 6 hours, the GSH1 enzyme activity of *apt1*
42 mutants was significantly lower than that of WT, but there was no significant difference
43 between *nacs*a mutants and WT. While our results show that under 15 min oxidative
44 stress, monomeric APT1 sense oxidative signal to form tetrameric APT1 (Fig. 3 a),
45 tetrameric APT1 can promote the depalmitoylation of NACsa and then significantly
46 up-regulate *GLYI* expression (Fig. 5 g), which leads to the increase of GSH/GSSG ratio
47 (compared with mutant lines) (Fig. 5 j). *glyI* mutant could affects ROS and GSH levels
48 even without stress (Fig. 5a and 5b). These results suggest that APT1-NACsa-GLYI
49 pathway can perceiving and regulating the redox balance in early stage of oxidative
50 stress, compared with GSH1.

51 Previous studies showed that the transcription level of *GSH1* increased after several
52 hours of stress treatment^{3,4}. Some studies showed that the transcription level of GSH1
53 did not significantly increase even after 24 h dark submergence low oxygen stress⁵. Our
54 study showed that the transcription level of *GSH1* did not significantly increase at stress
55 for 15 min, but after 6 h of oxidative stress treatment, the transcription level of *GSH1*
56 significantly increased, and the *apt1* mutant significantly decreased compared with WT,
57 and there was no significant difference between *nacs*a and the WT (Supplementary Fig.
58 9 b), which suggested that *GSH1* was not regulated by NACsa. There was no significant
59 difference of GSH1 enzyme activity in the wild type after oxidative stress 15 min,
60 compared with 0 h, which suggested that GSH1 had limited function in the early stage
61 of stress.

62 Thank you for your comments.

63

64 Supplementary Fig. 9

65 GSH1 enzyme activity (a) and relative *GSH1* expression (b) in WT, *pAPT1:APT1-*
66 *3×flag/apt1*, *apt1-1*, *apt1-2* mutant, *pNACsa:NACsa-GFP/nacs*a, *nacs*a-1 and *nacs*a-2
67 mutant lines after 0 min, 15 min, and 6 h of exposure to 4 mM H₂O₂ stress. The samples

68 comprised the whole root of the seedling. The mean values and SE were calculated
 69 from three independent experiments. The different words indicate a significant
 70 difference in different lines. (n=4, Tukey's nonparametric test, $P<0.05$).
 71

72

73 Figure 3

74 Oxidative stress promotes APT1 tetramer formation *in vivo*.

75

76

77 Figure 5

78 Relative *GLYI* relative expression (**g**) and GSH/GSSG ratio (**j**) in WT, *apt1*, *nacs1*, and
 79 *gly1* lines after 0 h, 15 min, and 6 h of exposure to 4 mM H₂O₂ stress.

80

81 Supplementary Methods

82 One hundred milligrams of frozen plant tissue was homogenized with the addition of 1
 83 mL of 0.1% (w/v) TCA, which was kept on ice at all times. Then, the sample was mixed
 84 well by vortexing and centrifuged at 15,000×g for 15 min at 4°C. Then, 2 mL tubes
 85 were prepared to perform the reaction by adding 0.5 mL of the supernatant and 0.5 mL
 86 of 10 mM phosphate buffer (pH 7.0). For the Blank, 0.5 mL of 0.1% (w/v) TCA was
 87 used instead of the supernatant. Then, 1 mL of potassium iodide was added to initiate
 88 the reaction as quickly as possible. The sample was mixed gently and left in the dark
 89 for 10 min. Then, 300 μL of the reaction mixture was transferred to a 96-well plate, and
 90 the absorbance at 390 nm was measured in a Multimode microplate reader (Spark®),

91 Tecan, Switzerland). The concentration and content of hydrogen peroxide in plant
92 tissue were calculated using a standard curve¹.

93 Determination of the H₂O₂ level by CM-H₂DCFDA

94 Control and treated tissues were harvested, and nearly 100 mg of tissue was ground in
95 liquid nitrogen. The ground tissue powder was placed in a preweighed 2 mL tube with
96 1 mL of 10 mM Tris-HCl (pH 7.2). The sample was centrifuged at 12,000×g for 20 min
97 at 4°C. Then, the supernatant was transferred to a fresh 2 mL tube. The instrument was
98 blanked with 10 mM Tris-HCl (pH 7.2). One hundred microlitres of supernatant was
99 diluted with 900 µL of 10 mM Tris-HCl (pH 7.2). Then, 10 µL of 1 mM CM-H₂DCFDA
100 (final concentration will be 10 µM) (C6827, Thermo) was added to the first sample, and
101 after 1 min, 10 µL of CM-H₂DCFDA was added to the next sample. The rest of the
102 samples were vortexed and incubated in the dark for 10 min. The fluorescence values
103 of samples treated with CM-H₂DCFDA were measured by a fluorometer (RF-5301 PC,
104 Shimadzu, Japan). The control was set by adding 100 µL of plant extract + 900 µL of
105 Tris-HCl (pH 7.2), inverting and mixing (using parafilm), and the values were read in
106 a fluorometer. This background fluorescence value was deducted from all readings. The
107 protein concentration in all samples was estimated using Bradford reagent (P0010,
108 Beyotime, China), and ROS levels are expressed as relative fluorescence units/mg of
109 protein extract. In this experiment, to rule out doubts about CM-H₂DCFDA not being
110 completely specific to ROS, we performed measurements on equal aliquots with
111 catalase (300 U/mL) added to one sample and repeated the measurement 6 times,
112 subtracting the catalase-insensitive background from the experimental value².

113

114

115 Supplementary references

- 116 1. Antoniou, C., Savvides, A., Georgiadou, E.C., Fotopoulos, V. (2018).
117 Spectrophotometric quantification of reactive oxygen, nitrogen and sulfur species
118 in plant samples. In: Alcázar, R., Tiburcio, A. (eds) Polyamines. Methods in
119 Molecular Biology, vol 1694. Humana Press, New York, NY.
120 https://doi.org/10.1007/978-1-4939-7398-9_16.
- 121 2. Jambunathan, N. (2010). Determination and detection of reactive oxygen species
122 (ROS), lipid peroxidation, and electrolyte leakage in plants. In: Sunkar, R. (eds)
123 Plant Stress Tolerance. Methods in Molecular Biology, vol 639. Humana Press.
124 https://doi.org/10.1007/978-1-60761-702-0_18.
- 125 3. Aya Hatano-Iwasaki, Ken'ichi Ogawa. (2012) Overexpression of *GSH1* gene
126 mimics transcriptional response to low temperature during seed vernalization
127 treatment of *Arabidopsis*. *Plant and Cell Physiology*, **53(7)**, 1195–1203,
128 <https://doi.org/10.1093/pcp/pcs075>.
- 129 4. Han, Y., Fan, T., Zhu, X. *et al.* (2019). WRKY12 represses *GSH1* expression to
130 negatively regulate cadmium tolerance in *Arabidopsis*. *Plant Mol Biol* **99**, 149–159.
131 <https://doi.org/10.1007/s11103-018-0809-7>.
- 132 5. Li-Bing Y., Yang-Shuo D., Li-Juan X., Lu-Jun Y., Ying Z., Yong-Xia L., Yi-Cong

133 Y., Le X., Qin-Fang C., Shi X. (2017) Jasmonate Regulates plant responses to
134 postsubmergence reoxygenation through transcriptional activation of antioxidant
135 synthesis, *Plant Physiology*, **173(3)**, 1864–1880,
136 <https://doi.org/10.1104/pp.16.01803>.
137

REVIEWERS' COMMENTS

Reviewer #3 (Remarks to the Author):

This manuscript presents compelling evidence that acyl-protein thioesterase 1 (APT1) is a redox sensor. Data are presented showing that APT1 exists as a monomer in the absence of stress. Monomerization is favoured by protein S-glutathionylation at C20, C22 and 9 C37, which also inhibits enzyme activity. Conversely, under oxidative conditions, APT1 forms a tetramer and regains enzymatic functions. The tetrameric form of APT1 depalmitoylates S-acetylated NAC (NACsa) transcription factors causing relocation of NACsa to the nucleus. The data regarding changes in reduced glutathione/oxidized glutathione (GSH/GSSG) ratio are accurate as is the reported upregulation of glyoxalase I expression. I am less convinced by the interpretation and conclusions related to these data. The methods used to measure hydrogen peroxide and the activity of glutamate-cysteine ligase (GSH1) are not sufficiently accurate to draw effective conclusions regarding the precise mechanisms of oxidant and antioxidant regulation. The authors have made a robust defence of the methods used to measure hydrogen peroxide. However, their arguments do not counteract problems with the specificity or accuracy of the measurements. The values for hydrogen peroxide level are far too high to reflect the real levels of this oxidant in the cytosol, and must reflect values for the tissue as a whole, particularly the apoplast/cell wall compartment. The authors should at least acknowledge this point in the discussion section. Furthermore, while the methods used to analyse GSH1 activity are sufficient for measurements in animal cells, they do not provide accurate information in plant extracts because of high activities of phosphatases and similar enzymes. GSH1 activity can only be measured accurately in plant extracts by HPLC analysis of relative amounts of substrates and products. The data on GSH1 activity should therefore be removed from this manuscript. These issues do not prohibit publication of the manuscript but the conclusions regarding mechanisms should be much more circumspect and the discussion section should be modified accordingly. A more general discussion about possible mechanisms of regulation of oxidant and antioxidant systems to adjust cellular redox homeostasis would be more accurate and informative. Crucially, these small changes would not detract from the overall value of the work.

REVIEWER COMMENTS

Reviewer #3 (Remarks to the Author):

This manuscript presents compelling evidence that acyl-protein thioesterase 1 (APT1) is a redox sensor. Data are presented showing that APT1 exists as a monomer in the absence of stress. Monomerization is favoured by protein S-glutathionylation at C20, C22 and 9 C37, which also inhibits enzyme activity. Conversely, under oxidative conditions, APT1 forms a tetramer and regains enzymatic functions. The tetrameric form of APT1 depalmitoylates S-acetylated NAC (NACsa) transcription factors causing relocation of NACsa to the nucleus. The data regarding changes in reduced glutathione/oxidized glutathione (GSH/GSSG) ratio are accurate as is the reported upregulation of glyoxalase I expression. I am less convinced by the interpretation and conclusions related to these data. The methods used to measure hydrogen peroxide and the activity of glutamate-cysteine ligase (GSH1) are not sufficiently accurate to draw effective conclusions regarding the precise mechanisms of oxidant and antioxidant regulation. The authors have made a robust defence of the methods used to measure hydrogen peroxide. However, their arguments do not counteract problems with the specificity or accuracy of the measurements. The values for hydrogen peroxide level are far too high to reflect the real levels of this oxidant in the cytosol, and must reflect values for the tissue as a whole, particularly the apoplast/cell wall compartment. The authors should at least acknowledge this point in the discussion section. Furthermore, while the methods used to analyse GSH1 activity are sufficient for measurements in animal cells, they do not provide accurate information in plant extracts because of high activities of phosphatases and similar enzymes. GSH1 activity can only be measured accurately in plant extracts by HPLC analysis of relative amounts of substrates and products. The data on GSH1 activity should therefore be removed from this manuscript. These issues do not prohibit publication of the manuscript but the conclusions regarding mechanisms should be much more circumspect and the discussion section should be modified accordingly. A more general discussion about possible mechanisms of regulation of oxidant and antioxidant systems to adjust cellular redox homeostasis would be more accurate and informative. Crucially, these small changes would not detract from the overall value of the work.

Thank you for your comments. We have used different methods to detect the level of H₂O₂ under drought stress (Supplementary Fig. 1 a-c) in order to determine the concentration of H₂O₂ that needs to be added to stimulate oxidative stress. With a ratio of extracellular to intracellular H₂O₂ concentrations of 100:1 (Sies and Jones, 2020; Noctor and Foyer, 2016), it is necessary to clarify that the detected H₂O₂ level is closer to the extracellular H₂O₂ concentration rather than the intracellular H₂O₂ concentration. The treatment of oxidative

stress adopted in this study was based on previous research on *Arabidopsis* (Wu et al. 2020). In the previous study, *Arabidopsis* was treated with 4 mM H₂O₂, and the concentration of 4 mM is similar to the results obtained in our study. Therefore, we adopted 4 mM H₂O₂ as the oxidative stress treatment in our study. We did not detect the actual concentration of H₂O₂ in the cytoplasm but used fluorescent probes to show that the intracellular H₂O₂ level in wild-type plants was significantly increased after exposure to oxidative stress induced by 4 mM H₂O₂ (Fig. 5a). The findings suggest that the level of intracellular oxidative stress increased after 4 mM H₂O₂ treatment. These results indicate that the application of 4 mM H₂O₂ externally can induce its transport to the cytoplasm through aquaporins or other transporters (Sies, et al. 2017; Waszczak et al. 2018), which prompts intracellular redox sensors to function.

In addition, we also deleted the relevant content of GSH1 and further discuss the regulatory mechanism of the redox balance and the role played by the APT1-NACsa-GLYI system in the Discussion (line 288-316). Thank you again for your comments.

References

- Sies, H., and Jones, D.P. (2020). Reactive oxygen species (ROS) as pleiotropic physiological signalling agents. *Nat. Rev. Mol. Cell Biol.* 21, 363-383.
- Noctor, G., and Foyer, C.H. (2016). Intracellular redox compartmentation and ROS-related communication in regulation and signaling. *Plant Physiol.* 171, 1581-1592.
- Wu, F., Chi, Y., Jiang, Z., Xu, Y., Xie, L., Huang, F., Wan, D., Ni, J., Yuan, F., and Wu, X. (2020). Hydrogen peroxide sensor HPCA1 is an LRR receptor kinase in *Arabidopsis*. *Nature* 578, 577-581.
- Sies, H., Berndt, C. & Jones, D. P. (2017). Oxidative stress. *Annu Rev Biochem.* 86, 715-748.
- Waszczak, C., Carmody, M. & Kangasjarvi, J. (2018). Reactive oxygen species in plant signaling. *Annu Rev Plant Biol.* 69, 209-236.